# Earthquake forecasting from paleoseismic records

Ting Wang [1] ✉, Jonathan D. Griffin [2], Marco Brenna [3], David Fletcher[4], Jiaxu Zeng[5], Mark Stirling[3], Peter W. Dillingham [1,6] & Jie Kang [7]

Forecasting large earthquakes along active faults is of critical importance for seismic hazard assessment. Statistical models of recurrence intervals based on compilations of paleoseismic data provide a forecasting tool. Here we compare five models and use Bayesian model-averaging to produce time-dependent, probabilistic forecasts of large earthquakes along 93 fault segments worldwide. This approach allows better use of the measurement errors associated with paleoseismic records and accounts for the uncertainty around model choice. Our results indicate that although the majority of fault segments (65/93) in the catalogue favour a single best model, 28 benefit from a model-averaging approach. We provide earthquake rupture probabilities for the next 50 years and forecast the occurrence times of the next rupture for all the fault segments. Our findings suggest that there is no universal model for large earthquake recurrence, and an ensemble forecasting approach is desirable when dealing with paleoseismic records with few data points and large measurement errors.

Large earthquakes are one of the most devastating natural hazards, not only because they cause a significant number of casualties, but also because they lead to widespread infrastructure damage. Researchers from multi-disciplinary backgrounds have been endeavouring to understand their underlying mechanisms and forecast their occurrences. Elastic rebound theory[1] forms the basis of the standard earthquake cycle of strain accumulation and release. This theoretical basis, combined with analysis of long-term earthquake records, supports the proposition that the recurrence times of large earthquakes along the same fault can be modelled with renewal processes[2–6]. A renewal process is a statistical model that describes event occurrences in time, treating each new occurrence as a renewal in which the system is reset. It assumes that the times between events (the inter-event time, such as the time between two consecutive earthquake occurrences) are independent and identically distributed. Probabilistic Seismic Hazard Analysis[7] frequently uses the Poisson process, which is a special type of renewal process that describes events effectively occurring randomly in time. Recent analysis of global paleoseismic records nevertheless suggests that large earthquake recurrence on individual fault segments is typically more periodic than expected from a Poisson process[8–10]. Such periodic patterns can be captured by other renewal processes, such as the Gamma renewal process. An alternative to renewal processes for modelling paleoseismic records uses a Long-Term Fault Memory Model[11], which assumes that the timing of future events is dependent on not only the time elapsed since the most recent event (as in renewal models) but also the previous inter-event times. However, when considering global dataset and synthetic earthquake records there appears to be no significant correlation or anti-correlation between successive inter-event times for the vast majority of the earthquake records[9,12]. Therefore, in this study we only focus on renewal processes.

[1]Department of Mathematics and Statistics, University of Otago, Dunedin 9016, New Zealand. [2]Community Safety Branch, Geoscience Australia, Symonston 2609 ACT, Australia. [3]Department of Geology, University of Otago, Dunedin 9016, New Zealand. [4]David Fletcher Consulting Limited, 67 Stornoway Street, Karitane 9471, New Zealand. [5]Department of Preventive and Social Medicine, Otago Medical School, University of Otago, Dunedin 9016, New Zealand. [6]Coastal People: Southern Skies Centre of Research Excellence, University of Otago, Dunedin 9016, New Zealand. [7]Beef + Lamb New Zealand Genetics, 3 Crawford Street, PO Box 5501, Dunedin 9054, New Zealand. ✉ e-mail: ting.wang@otago.ac.nz

Previous earthquake forecasts using paleoseismic data were mainly based on one single best-model selected from a number of candidate models according to an information criterion, such as the Akaike Information Criterion (AIC[13]). This approach ignores the fact that there can be model uncertainty, i.e. uncertainty as to which model is best. If the AIC values for several models are not very different, it may be better to use a combination of the forecasts from those models, rather than rely solely on the forecast from the model with the lowest AIC. In real-world applications such model uncertainty may have a dominant effect over uncertainty in the estimates of the parameters in each model[14,15]. This becomes a serious limitation when the sample size is relatively small (only 14 out of 93 earthquake records considered in this study have more than 10 events). Failing to acknowledge model uncertainty by selecting a single best model may produce erroneous or unrealistically precise forecasts.

In this study, we use five candidate renewal processes to investigate the recurrence patterns of large earthquakes using paleoseismic records from 93 worldwide fault segments previously compiled in Griffin et al.[9] and supplemented with additional records from other publications (See Data availability section). We use Bayesian model-averaging to carry out probabilistic forecasts of future large earthquakes to account for model uncertainty. We compare the forecasts from each single model with that from model averaging. We find that there is no single best model that universally describes the recurrence of large earthquakes for the 93 fault segments considered here, nor for fault segments with the same faulting styles, from the same tectonic region, or even within the same fault system. We provide the distribution of the probability that at least one large earthquake will occur in the next 50 years along each fault segment. We also carry out a leave-one-out test to compare the performance of the model-averaged forecast and the Poisson forecast, as the latter is frequently used in national hazard models[7].

## Results

### No universal model for large earthquakes

Modelling of large-earthquake recurrence times from paleoseismic records typically either uses the BPT renewal process, because of its physical explanation of the earthquake process[4,6,16], or selects the best model among a few candidate renewal processes, based on an information criterion[3,17]. As discussed above, inference based on a single best model ignores potential model uncertainty.

In order to minimise the impact of model uncertainty we use a model-averaging approach and consider geological and historical records of large-earthquake occurrence times from 93 worldwide fault segments (Table 1). Here, we consider a "fault segment" as a section of a fault that is recognised as being geometrically distinct, and that is likely to define the boundaries of at least some earthquake ruptures on the fault. Some models, such as the third Uniform California Earthquake Rupture Forecast (UCERF3[16]) and the 2022 New Zealand National Seismic Hazard Model[18], relax the strict fault segmentation assumption in their earthquake probability model component by considering interactions between different fault subsections. Nevertheless, for each individual fault subsection the fundamental elastic-rebound theory part of the model uses a Brownian passage-time (BPT) renewal model. Here we focus on the elastic-rebound theory part of the earthquake probability model component, and thus do not consider interactions between fault segments. Once this layer of modelling is improved, interactions between subsections can be considered in order to forecast large earthquakes that may rupture multiple fault subsections and/or faults. Improved forecasts from this layer will strengthen a holistic hazard model that includes fault models, deformation models, earthquake rate models, and earthquake probability models.

We select fault segments with records of at least five large earthquakes in order to be able to fit two-parameter models to the inter-

event times. We only consider the occurrence times of earthquakes that left detectable geological evidence[9]. Earthquakes that leave a geological signature are assumed to be large and significant for seismic hazard, but we do not explicitly consider their magnitudes, or the magnitude distribution (i.e. characteristic vs Gutenberg-Richter) in our model. The number of large earthquakes in the paleoseismic record of each fault segment is small, with a maximum of 35 events for the Chile Megathrust[19], only 3 fault segments having more than 20 events, and 14 having more than 10 events. The measurement errors associated with most dated earthquake ages are large, resulting in $1\sigma$ uncertainties of $25 \pm 12\%$ for inter-event times in the data considered here. To maximise the reliability from the dataset of all event ages and to make good use of the measurement errors, we simulate 100 Monte Carlo (MC) samples from the empirical distribution of the occurrence times for each fault segment provided in the literature. If any earthquake has more than one dated age published in the literature, we take the average of the different age distributions and sample from this averaged distribution.

The 100 MC samples of occurrence times for each fault segment form each dataset. We fit five different renewal processes to each dataset, including the Poisson process, and the Gamma, Weibull, BPT and lognormal renewal processes (see Methods). In each model, we allow the parameters to vary for different MC samples from the same fault, which captures the similarities between the 100 MC samples. We use a Markov Chain Monte Carlo (MCMC) algorithm to generate samples from the joint posterior distribution of all the model parameters for each model. These posterior samples of parameters are then used to simulate future earthquake occurrence times from each fault, thereby providing us with forecasts from each model.

For each model fitted to each fault segment, we calculate the Watanabe-Akaike Information Criterion (WAIC[20]), a measure of the prediction performance of a model in Bayesian analysis. For model-averaging, we calculate the WAIC weight[15] for each model for all 93 fault segments (see Methods), as shown in Fig. 1a. Model-averaged forecasts of future earthquakes for each fault are obtained by combining the forecasts from each individual model using these WAIC weights (see Methods). For a fault segment with any single model having weight ≥0.95, the model-averaged forecast is very close to the forecast from that model (i.e., the single-best model), as we would expect. The Weibull model best fits 41 (44.1%) fault segments, the Gamma model best fits 5 (5.4%) fault segments, the BPT model best fits 4 (4.3%) fault segments, and the lognormal model best fits 15 (16.1%) fault segments. The Poisson model has no weight greater than 0.7. The remaining 28 (30.1%) fault segments have weights < 0.95 for each single model, suggesting that model-averaging will be most beneficial for those fault segments.

WAIC weights provide a numerical comparison of the amount by which a model is better at prediction than another, as they show how much weight should be given to the prediction from each of these models when calculating a model-averaged prediction. Based on the WAIC weights in Fig. 1a, we can see that the predictive performance of the Weibull model is not uniformly better than the others. For 33 fault segments, the WAIC weight for the Weibull model is close to 0, which suggests that for these fault segments the estimated predictive accuracy of the other models is much better than that of the Weibull model. Among the 11 San Andreas Fault segments we find three different best-fit models. Although they all suggest that every segment exhibits quasi-periodic behaviour (see discussion later), segments with larger sample sizes were best fit by a Weibull model. It is unclear if this is simply an outcome of sampling or due to real differences in recurrence behaviour between the different segments, including how neighbouring fault segments interact. Variability in observed earthquake recurrence behaviour at paleoseismic sites on the San Andreas Fault has been proposed to be (at least partially) due to overlap of ruptures occurring on neighbouring segments[21] and this has been supported by studies

**Table 1 | Probability of at least one event occurring in the next 50 years from the 93 fault segments**

| ID | Fault name | N | MA forecast prob (%) | BM | BM forecast prob (%) |
|----|-----------|---|---------------------|----|---------------------|
| 1 | Alaska PWS Copper | 9 | 0.001(~0,0.006) | W | 0.001(~0,0.006) |
| 2 | Alpine Hokuri Ck South Westland | 25 | 26.898(22.22,32.492) | W | 26.898(22.22,32.492) |
| 3 | Awatere East | 10 | 2.806(2.382,3.254) | W | 2.806(2.382,3.254) |
| 4 | Bree | 6 | 0.042(0.008,0.121) | W | 0.042(0.008,0.121) |
| 5 | Cadell | 9 | 0.038(0.034,0.043) | L | 0.038(0.034,0.043) |
| 6 | Cascadia | 19 | 9.751(7.763,11.329) | W | 9.751(7.763,11.329) |
| 7 | Cascadia Nth | 10 | 6.884(5.369,7.84) | W | 6.884(5.369,7.84) |
| 8 | Cascadia Sth | 19 | 9.676(8.502,10.873) | G | 9.676(8.502,10.873) |
| 9 | Chile Margin | 35 | 30.379(29.116,31.673) | W | 30.379(29.116,31.673) |
| 10 | Cloudy Fault | 6 | 1.618(1.372,1.893) | W | 1.618(1.372,1.893) |
| 11 | Daqingshan Piedmont Hohhot | 7 | 6.136(5.316,6.933) | G | 6.142(5.423,6.923) |
| 12 | Dead Sea Beteiha | 11 | 20.336(18.506,22.139) | P | 20.424(18.841,22.101) |
| 13 | Dead Sea Jordan | 12 | 4.35(3.941,4.798) | W | 4.336(3.932,4.777) |
| 14 | Dead Sea Qatar | 10 | 9.999(8.886,11.035) | P | 10.095(9.214,11.035) |
| 15 | Dead Sea Taybeh | 11 | 6.91(6.234,7.653) | L | 6.91(6.234,7.653) |
| 16 | Dead Sea Yammouneh | 12 | 5.61(4.583,6.942) | W | 5.61(4.583,6.942) |
| 17 | Dunstan | 6 | 0.567 (0.465, 0.685) | L | 0.567 (0.465, 0.685) |
| 18 | East Kunlun Kusaihu | 10 | ~0(~0,~0) | L | ~0(~0,~0) |
| 19 | East Kunlun Xidatan | 6 | 3.286(2.866,3.758) | L | 3.286(2.866,3.758) |
| 20 | Elashan | 5 | 4.116(3.281,5.041) | L | 4.108(3.31,4.982) |
| 21 | Elsinore Temecula | 5 | 4.029(2.404,5.227) | L | 4.029(2.404,5.227) |
| 22 | Fatigue Wash | 6 | 0.042(0.027,0.061) | G | 0.04(0.026,0.058) |
| 23 | Futugawa | 5 | ~0(~0,~0) | G | ~0(~0,~0) |
| 24 | Garlock El Paso Peaks | 6 | 3.338(2.892,3.824) | W | 3.327(2.89,3.784) |
| 25 | Garlock Twin Lakes | 6 | 3.991(2.806,5.132) | W | 3.983(2.804,5.105) |
| 26 | Gulang Tianqiaogou | 5 | 3.022(2.518,3.67) | W | 3.019(2.517,3.666) |
| 27 | Haiyuan Middle | 7 | 9.301(6.504,13.88) | W | 9.301(6.504,13.88) |
| 28 | Hayward Tysons | 11 | 42.866(38.168,48.161) | W | 42.866(38.168,48.161) |
| 29 | Helanshan Eastern Piedmont | 5 | ~0(~0,0.001) | W | ~0(~0,0.001) |
| 30 | Hope | 6 | 19.518(13.297,21.579) | B | 19.672(17.759,21.652) |
| 31 | Hope Conway | 5 | 15.865(12.658,20.142) | L | 15.939(12.617,20.308) |
| 32 | Hyden | 5 | 0.062(0.045,0.078) | G | 0.064(0.046,0.08) |
| 33 | Irpinia | 5 | ~0(~0,0.245) | W | ~0(~0,0.245) |
| 34 | Javon Canyon | 5 | 5.855(2.885,8.982) | W | 5.855(2.885,8.982) |
| 35 | Kiri | 5 | 5.931(4.427,7.59) | L | 5.911(4.473,7.475) |
| 36 | Lachlan | 5 | 3.756(3.034,4.493) | W | 3.756(3.034,4.493) |
| 37 | Lake Edgar | 5 | 0.023(0.019,0.028) | G | 0.023(0.019,0.028) |
| 38 | Langshan Piedmont Xibulong East | 5 | 1.929(1.618,2.295) | L | 1.929(1.618,2.295) |
| 39 | Lenglongling | 5 | 0.006(~0,0.041) | L | 0.006(0.001,0.041) |
| 40 | Mangatete | 6 | 0.032(0.001,0.189) | W | 0.032(0.001,0.189) |
| 41 | Nankai Trough | 8 | 13.916(11.782,16.187) | W | 13.916(11.782,16.187) |
| 42 | New Guinea | 6 | 4.868(4.173,5.945) | W | 4.826(4.157,5.561) |
| 43 | North Anatolian Cukurcimen | 6 | 1.023(0.619,1.585) | W | 1.023(0.619,1.585) |
| 44 | North Anatolian Elmacik | 8 | 1.335(0.533,1.945) | G | 1.365(0.922,1.96) |
| 45 | North Anatolian Gunalan | 6 | 0.139(0.046,0.37) | B | 0.139(0.046,0.368) |
| 46 | North Anatolian Kavakkoy | 6 | 14.103(11.425,16.896) | B | 14.103(11.425,16.897) |
| 47 | North Anatolian Lake Ladik | 7 | 21.196(5.798,34.833) | W | 21.196(5.798,34.833) |
| 48 | North Anatolian Yaylabeli | 5 | 4.885(2.525,7.957) | B | 4.886(2.506,7.987) |
| 49 | Okaya | 9 | 1.992(0.983,3.319) | W | 1.992(0.983,3.319) |
| 50 | Paeroa | 7 | 1.225(0.962,1.493) | W | 1.219(0.961,1.471) |
| 51 | Pasuruan | 5 | 17.776(15.402,20.38) | W | 17.771(15.411,20.365) |
| 52 | Pihama | 7 | 0.337(~0,0.589) | L | 0.338(~0,0.589) |
| 53 | Porters Pass East | 6 | 2.14(1.78,2.511) | W | 2.14(1.78,2.511) |
| 54 | Qilianshan Laohushan | 8 | ~0(~0,~0) | B | ~0(~0,~0) |
| 55 | Rangipo | 7 | 2.518(0.503,4.55) | G | 2.518(0.503,4.55) |

**Table 1 (continued) | Probability of at least one event occurring in the next 50 years from the 93 fault segments**

| ID | Fault name | N | MA forecast prob (%) | BM | BM forecast prob (%) |
|----|-----------|---|---------------------|----|--------------------| 
| 56 | Reelfoot | 5 | 2.192(1.569,2.851) | L | 2.193(1.57,2.851) |
| 57 | Rocky Valley | 5 | 0.081(0.051,0.109) | W | 0.081(0.053,0.108) |
| 58 | Rotoitipakau | 9 | 3.093(2.537,3.73) | W | 3.081(2.532,3.677) |
| 59 | San Andreas Big Bend | 10 | 40.848(36.147,45.931) | W | 40.848(36.147,45.931) |
| 60 | San Andreas Burro | 7 | 34.498(30.194,39.242) | W | 34.498(30.194,39.242) |
| 61 | San Andreas Carrizo | 6 | 59.954(51.174,71.794) | W | 63.604(54.216,73.246) |
| 62 | San Andreas Coachella | 7 | 24.399(19.926,30.01) | L | 24.399(19.926,30.01) |
| 63 | San Andreas Mendocino | 5 | 0.14(0.031,0.441) | L | 0.14(0.031,0.44) |
| 64 | San Andreas Mission Ck | 5 | 46.826(37.066,58.07) | W | 46.879(37.486,58.098) |
| 65 | San Andreas Pallet Ck | 9 | 45.121(40.419,50.149) | W | 45.121(40.419,50.149) |
| 66 | San Andreas Pittman | 7 | 38.991(32.317,47.317) | W | 38.994(32.319,47.32) |
| 67 | San Andreas Thousand Palms | 5 | 40.079(33.627,58.286) | G | 39.026(33.33,44.753) |
| 68 | San Andreas Vedanta | 12 | 17.033(15.696,18.432) | W | 17.033(15.696,18.432) |
| 69 | San Andreas Wrightwood | 15 | 70.727(64.942,76.399) | W | 70.727(64.942,76.399) |
| 70 | San Jacinto HogLake | 21 | 24.749(23.397,26.153) | W | 24.749(23.397,26.153) |
| 71 | San Jacinto Mystic Lake | 13 | 46.015(43.077,48.948) | G | 46.015(43.077,48.948) |
| 72 | Serteng Piedmont Wujia | 5 | 0.88(0.736,1.046) | W | 0.879(0.735,1.045) |
| 73 | Snowden | 5 | 0.388(0.156,0.658) | W | 0.388(0.156,0.658) |
| 74 | Solitario Canyon | 5 | 0.041(0.033,0.048) | L | 0.041(0.033,0.048) |
| 75 | Stagecoach Road | 5 | 0.114(0.078,0.155) | L | 0.126(0.091,0.16) |
| 76 | Sumatra Mentawai | 13 | 58.364(54.815,65.233) | G | 58.101(54.693,61.437) |
| 77 | Tanna | 9 | 1.903(1.251,2.686) | W | 1.903(1.251,2.686) |
| 78 | Teton Lakes | 8 | 2.757(2.387,3.173) | L | 2.757(2.387,3.173) |
| 79 | Vernon | 5 | 2.188(1.338,3.684) | W | 2.188(1.338,3.684) |
| 80 | Wairarapa South | 6 | 0.029(0.013,0.06) | W | 0.029(0.013,0.06) |
| 81 | Wairau | 5 | 10.68(6.813,17.663) | L | 10.329(6.609,17.281) |
| 82 | Waitangi | 7 | 0.21(0.128,0.289) | B | 0.21(0.128,0.289) |
| 83 | Wasatch Brigham | 6 | 14.691(12.395,17.373) | L | 14.663(12.38,17.302) |
| 84 | Wasatch Nilphi | 6 | 5.199(4.085,6.607) | W | 5.199(4.085,6.607) |
| 85 | Wasatch Weber | 5 | 2.816(1.627,3.598) | W | 2.816(1.627,3.598) |
| 86 | Wharekuri | 5 | 0.137(0.116,0.161) | G | 0.138(0.116,0.162) |
| 87 | Whirinaki | 8 | 0.643(0.479,0.828) | G | 0.633(0.471,0.823) |
| 88 | Windy Wash | 7 | 0.005(0.001,0.01) | W | 0.006(0.003,0.01) |
| 89 | Wulashan Piedmont Heshunzhuang Botou | 7 | 2.261(1.924,2.655) | W | 2.261(1.924,2.655) |
| 90 | Wulashan Piedmont Jinmiaozi Heshuzhuang | 6 | 1.193(1.006,1.388) | L | 1.193(1.006,1.388) |
| 91 | Wutai North Piedmont | 5 | 2.46(2.056,2.958) | W | 2.393(2.032,2.77) |
| 92 | Xorkoli | 9 | 8.037(6.235,9.312) | W | 8.037(6.235,9.312) |
| 93 | Zemuhe | 8 | 0.007(0.003,0.017) | W | 0.007(0.003,0.017) |

Forecast probabilities: median (95% credible interval). *N* number of events in the paleoseismic records, MA model-averaged, BM Single best model, i.e. with the largest WAIC weight.

using earthquake simulators[22]. In contrast, relatively strong quasi-periodic recurrence on the Alpine Fault has been attributed to its geometric simplicity and relative isolation from other faults[23], although its fault geometry has also been invoked to explain the variability in earthquake inter-event times[24]. The persistence (or otherwise) of rupture barriers between segments[25] may also be a significant factor controlling the distribution of inter-event times observed on a fault segment. Therefore, because of the limitations of the available data, even within a single well-studied fault system, we cannot use a universal single best model, and it is not clear that one exists.

Best-model frequencies in Fig. 1b, c indicate that the Weibull renewal process is more likely to best reproduce recurrence times for fault segments with longer records compared to the BPT renewal process. The fault segments that have the Weibull renewal process as the single-best model have on average about 1.59 (95% CI 1.10–2.47) times more large earthquakes than those for which the BPT renewal

process is the single-best model. The fault segments that have the BPT renewal process as the single-best model all have fewer than 10 events in the paleoseismic records. When we removed the last event in each record, in order to carry out retrospective forecasts (see section Assessment of Prediction Error), the fault segments with more than 15 events all have the Weibull renewal process as the single-best model; only one fault segment with 12 events has the BPT renewal process as the single-best model. It is therefore difficult to determine whether the dependence of the favoured model on sample size is due to real differences in the best model or simply an outcome of the small amount of data.

Previous studies[26–28] have shown that the standard deviations of the scaled (divided by the mean value) inter-event times along several fault segments appear to be constant. For each of the 93 fault segments, we calculated the scaled inter-event times for each MC sample, and then reported the median and the 2.5% and 97.5% quantiles of the standard deviations of the scaled inter-event times calculated for the

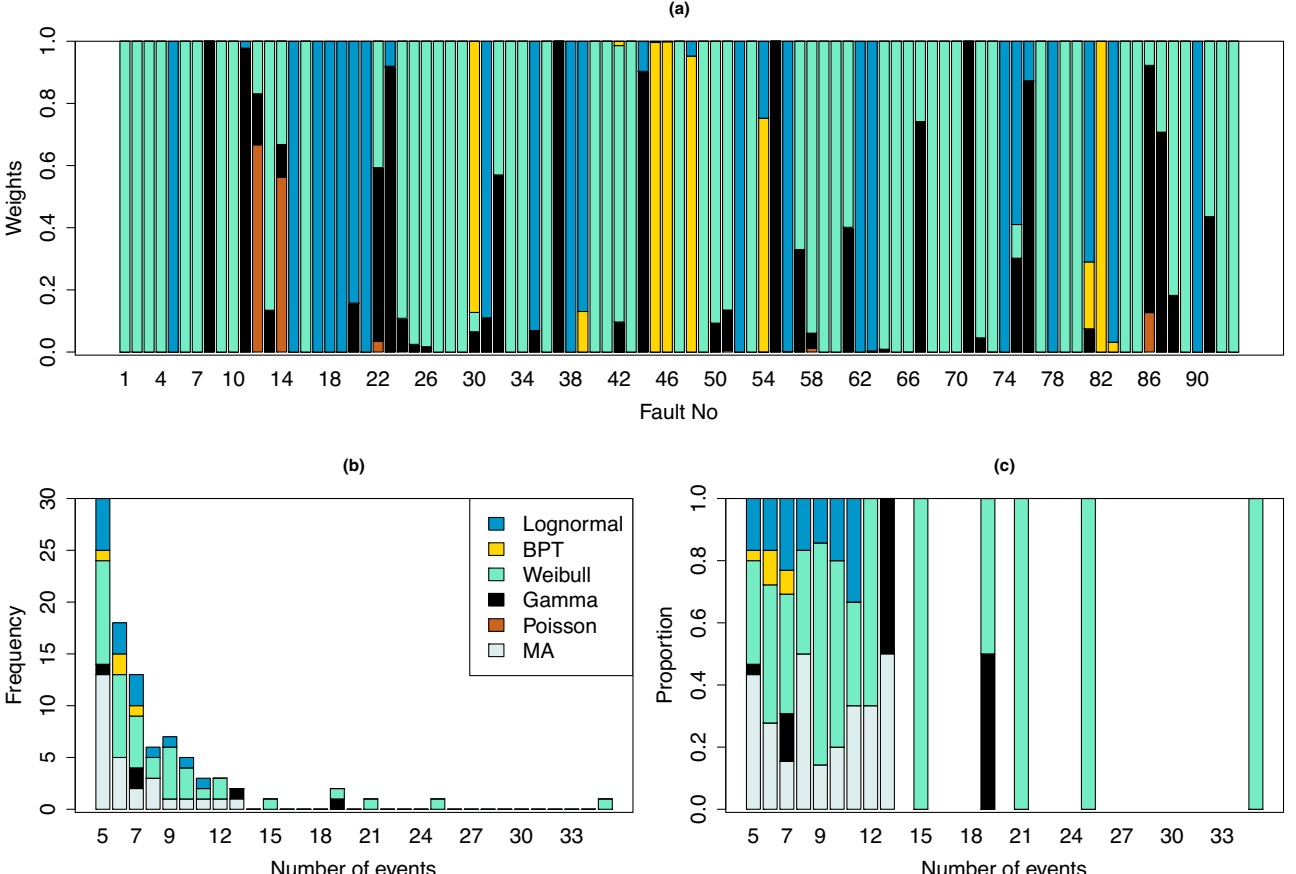

**Fig. 1 | The Watanabe-Akaike Information Criterion (WAIC) weight for each model for all 93 fault segments. a** Model weights for each of the 93 fault segments. Frequency (**b**) and proportions (**c**) of preferred models against number of events recorded at a fault segment. BPT Brownian passage-time, MA model-averaging.

100 MC samples. Fault segments with higher rates of earthquake occurrence appear to have smaller standard deviations of the scaled inter-event times (Supplementary Fig. 1 (e)). The median standard deviations of the scaled inter-event times for the 41 fault segments that were best fit by a Weibull model are all smaller than 0.8 except for one case. In contrast, more than half of the median standard deviations of the scaled inter-event times for the 15 fault segments that were best fit by a lognormal model are larger than 0.8 (Supplementary Fig. 1a, d). The large median standard deviations are related to fault segments that have inter-event times with long-tailed distributions which result in a large spread of data values. These long-tailed distributions are best fit by a lognormal renewal process.

**Probabilistic forecasting**

The estimated probability that at least one large earthquake will occur in the next 50 years along each fault segment is shown in Fig. 2 and Table 1. The highest probability of a large earthquake in the next 50 years appears to be along the Carrizo and Wrightwood segments of the San Andreas Fault, and the Mentawai Segment of the Sumatra megathrust, with median probabilities over 50%. The upper limit of the 95% credible interval (CI) of this probability exceeds 75% for the Wrightwood segment of the San Andreas Fault. These results are consistent with previous suggestions that these faults are in or approaching the beginning of a new seismic cycle[29–33]. Other sites with high probabilities of rupture (>20%) in the next 50 years include the Alpine Fault, the Chile Margin, Hayward Fault, most other segments of the San Andreas Fault, the San Jacinto Fault, and certain segments of the Dead Sea Transform and North Anatolian Fault (Table 1). If one fault segment has a high probability of rupture, it may affect the probability of rupture at a neighbouring fault segment. This is not built in our method

for the global data analysis, but can be considered for a particular local fault system. The lowest probabilities of a large earthquake in the next 50 years are along the Kusaihu segment of the East Kunlun Fault, the Futugawa Fault, the Eastern Piedmont segment of the Helanshan Fault, and the Laohushan segment of the QilianshanFault, with the upper limit of the 95% credible interval of these probabilities all being lower than 0.001%. On average, faults along a plate boundary are about 32 (95% CI 9–125) times more likely to have a large earthquake in the next 50 years than intraplate faults.

Model-averaging and best-model approaches can give significantly different forecast probabilities for an earthquake occurring within the next 50 years. The maximum difference in the forecast probability between the two approaches is about 14% for the Thousand Palms segment of the San Andreas Fault, where the model-averaging approach gives a higher probability than the best-model approach (Table 1). Having said that, for about 90% of the fault segments, this difference is less than 0.1%. When considering the forecast occurrence times, for the 28 fault segments where the single-best model has a WAIC weight less than 0.95, the two approaches give quite different results. The maximum difference between the credible intervals obtained using the model-averaging and single-best model approaches is over 10,000 years, and the difference is over 50 years for 43% of the 28 fault segments that benefited from a model-averaging approach. Figure 3 shows the forecast occurrence time of the next large earthquake along each fault using the model-averaging approach, while Table 1 lists the probabilities of a large earthquake occurring within the next 50 years. A comparison of the forecasts from different models is in the supplement (Supplementary Figs. 2–4).

A quantitative comparison of our probabilistic forecasts with those published in other studies is challenging (Supplementary Data 1).

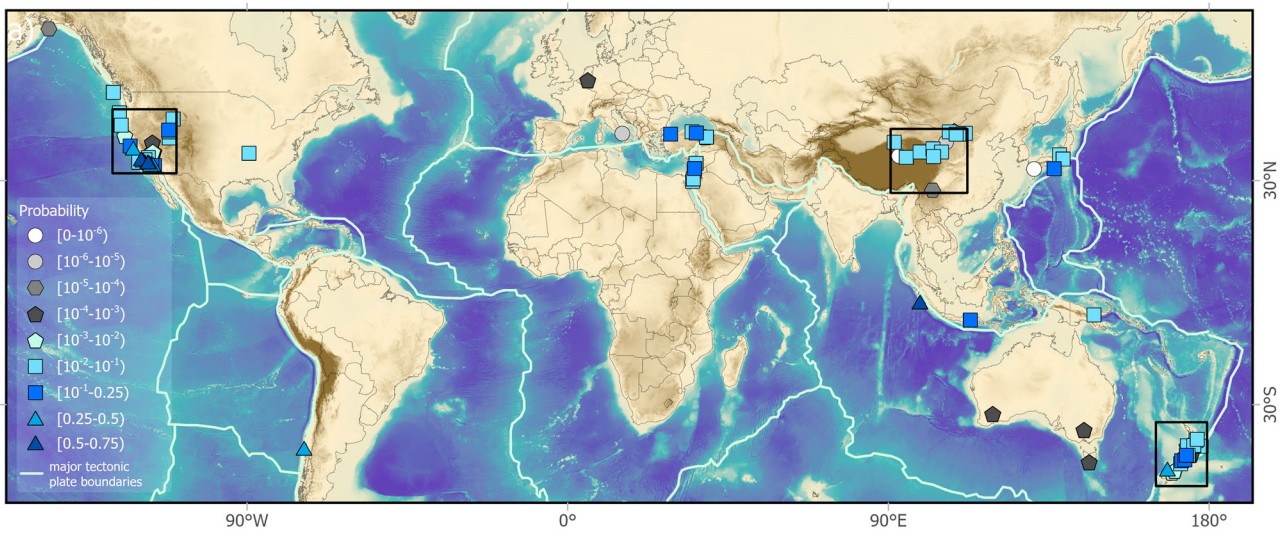

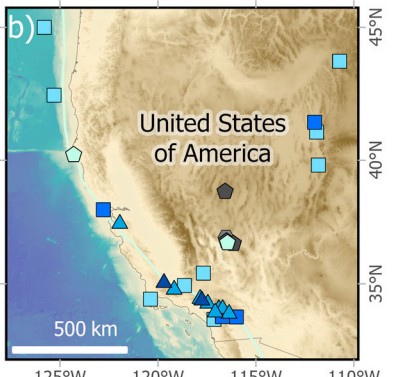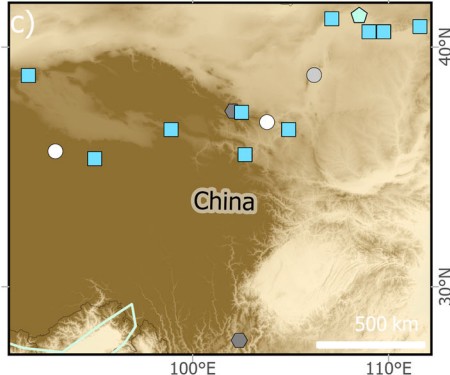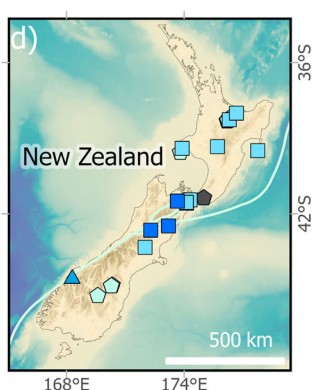

**Fig. 2 | Forecast probability of an event occurring in the next 50 years for the 93 fault segments.** The values are the medians of the posterior forecast probabilities. **a** World map; **b** San Andreas fault segments and surroundings; **c** central China; **d** New Zealand. Scale bars in **b**–**d** are approximate. ArcGIS software by Esri was used to create the map. Basemap data sources: ETOPO elevation model (ETOPO 2022, 60 Arc-Second Resolution, Bedrock elevation geotiff)[50], GNS Science, Natural Earth, USGS. Map projections are WGS 1984 Web Mercator (auxiliary sphere), WKID 1857.

Reports of the mean recurrence interval estimates are equivalent to forecasting the next occurrence time using a Poisson process (this comparison is in Supplementary Figs. 2–4). A meaningful comparison is difficult when the forecast of the probability of the next earthquake occurrence is specified in terms of a start date and a fixed time period, as these vary between studies. For example, UCERF3 provides 30 year probabilities from 2014 CE aggregated by parent fault section[16], whereas we provide 50 year probabilities from 2022 CE on each fault segment. The UCERF3 forecasts for the San Andreas fault segments all have similar mean values and overlapping ranges, whereas our forecast probabilities show greater variability between segments, with several non-overlapping credible intervals (Supplementary Fig. 5). We found previous forecasts for another eight fault segments in our study. All except two of our forecasts overlap within uncertainties with previous values, although our uncertainty bounds are typically narrower (Supplementary Fig. 5). Having said that, UCERF3 provides the ranges of forecast probabilities rather than 95% confidence intervals.

**Clustering or periodic behaviour**

For a Gamma or Weibull renewal process, if the estimated shape parameter $\alpha < 1$, then the process tends to have clustering behaviour, while if $\alpha > 1$, the process tends to have quasi-periodic behaviour, with larger $\alpha$ suggesting more periodic behaviour. For a BPT renewal process with probability density function as defined in Eq (3), smaller $\beta$

(for $\beta < 1$) suggests a more symmetrical probability density function of inter-event times and hence more periodic behaviour. A lognormal renewal process describes quasi-periodic behaviour, with a smaller standard deviation $\sigma$ corresponding to more periodic behaviour.

Based on the parameter estimates from the Gamma and Weibull renewal processes, five fault segments appear to show clustering behaviour, with the upper 95% credible limit of the shape parameter from each of these models being less than 1. These are Cadell, Dunstan, Lake Edgar, Solitario Canyon, and Waitangi, all of which have low earthquake occurrence rates. For Waitangi, the BPT renewal process has a WAIC weight over 0.95, and the estimates of $\beta$ are over 2.5, confirming a clustering behaviour. For Lake Edgar, the Gamma renewal process has a WAIC weight over 0.95. Six fault segments appear to demonstrate near Poisson behaviour, Dead Sea Beteiha, Dead Sea Qatar, Dead Sea Taybeh, Langshan Piedmont Xibulong East, Reelfoot, and Wharekuri, with the 95% credible interval of the shape parameter for both the Gamma and Weibull model containing 1. The remaining 82 fault segments show quasi-periodic behaviour. These results are consistent with a previous study[9] that used different tests to check quasi-periodic recurrence behaviour.

A regression model for the relationship between the shape parameter of a Weibull renewal process and the earthquake rate, tectonic setting, faulting type, and the number of earthquakes of each of the 93 fault segments (Fig. 4) suggests that when the

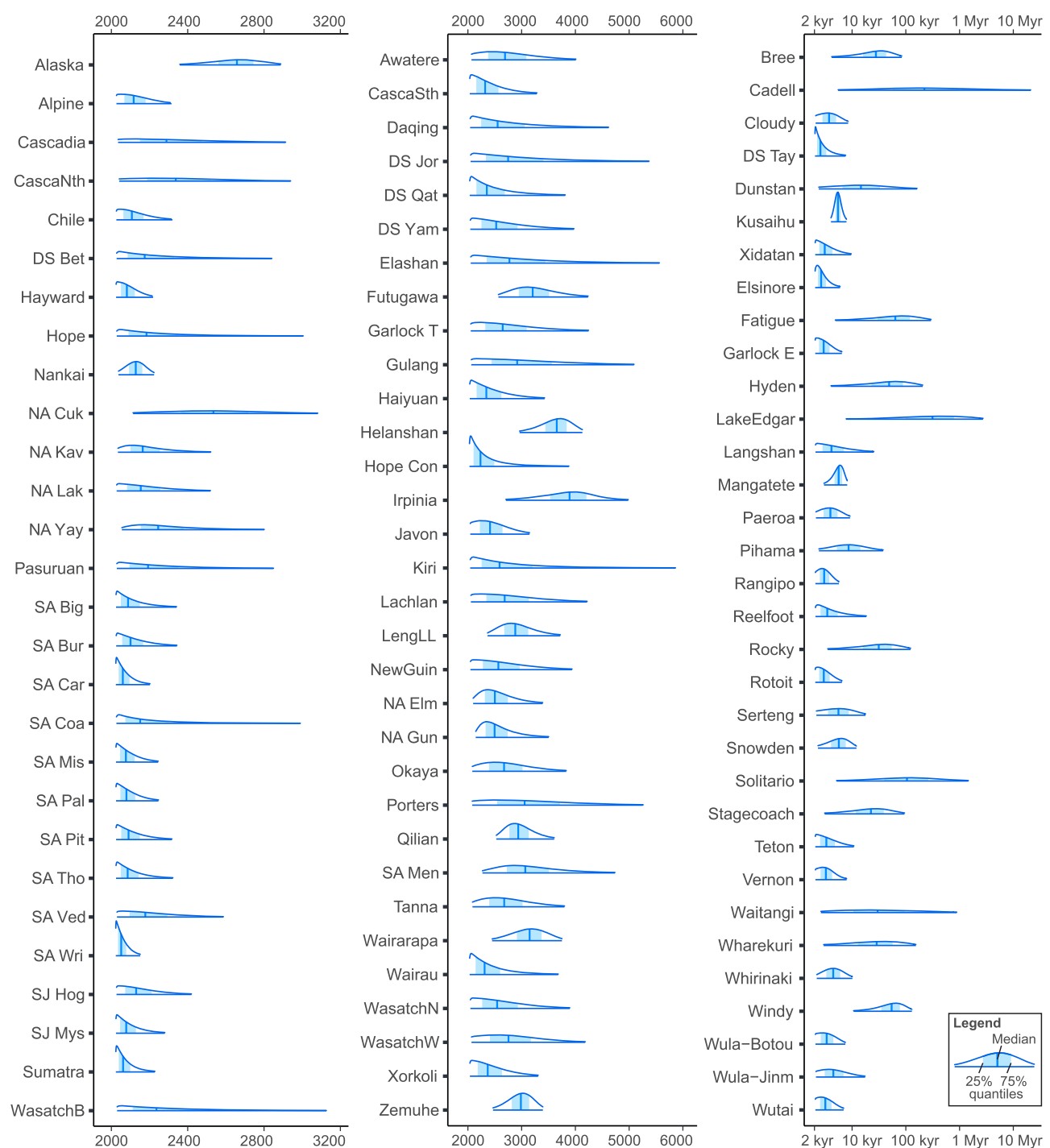

**Fig. 3 | Forecast occurrence times of the next large earthquake for the 93 fault segments.** The x-axis is in years CE.

earthquake rate increases 10 fold (e.g., from 0.0001 to 0.001), the shape parameter of the Weibull renewal process increases by about 24% (95% CI: 5%–45%). This suggests that fault segments with higher earthquake rates tend to have more periodic behaviour. The shape parameter of the Weibull renewal process for fault segments located in a stable continental intraplate setting is about 87% (95% CI: 51%–148%) of that for fault segments located at or near a plate boundary. Although the 95% CI is wide and covers 100%, the posterior density plot (Supplementary Fig. 6) suggests that there is a high probability (0.7) that the latter may be more periodic than the former. There are only five fault segments from a stable continental intraplate setting, so to reach a more robust conclusion, more data from intraplate fault segments are needed. These findings are consistent with and nuancing those from past studies[9,10]. The shape parameter of the Weibull renewal process for fault segments located in an active intraplate setting (predominantly faults in central China) is about 1.4 (95% CI: 1.1–1.9) times that for fault segments located at or near plate boundary, suggesting that the former appear to be more periodic than the latter. The shape parameter of the Weibull renewal process for fault segments located in a subduction region is about 1.9 (95% CI: 1.1–3.3) times that for fault segments located at or near plate boundary, suggesting that the former appear to be more periodic than the latter. On average, the Weibull shape

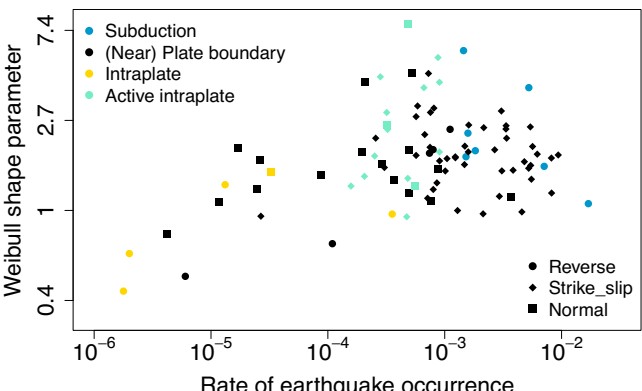

**Fig. 4 | Relationship between the Weibull shape parameters and the fault characteristics.** The x axis is earthquake rate per year (on log10 scale) estimated from a Poisson process. The y axis is the mean of the log shape parameter of the Weibull renewal process fitted to each fault segment. The larger the shape parameter, the more periodic the behaviour. Note that some of the pattern observed will be influenced by the number of earthquakes along each fault segment, which is not plotted.

parameter for reverse faults is about 69% (95% CI: 46%–104%) of that for normal faults, suggesting that large earthquakes recur more periodically on normal faults than on reverse faults. Note that obtaining long earthquake records for reverse faults is often more difficult than for normal faults, due to progressive burial of the evidence for previous earthquakes. With more data becoming available in the future, one could investigate whether variations in strain rate or kinematics have a significant impact on record length and event preservation.

The likelihood of systematically missing very short inter-event times in the paleoseismic records and hence biasing our analysis is low. If fault re-rupture commonly occurred shortly after previous earthquakes, then we would expect to see this frequently in the historical record of surface rupturing earthquakes, which we largely do not[34]. While preservation of paleoseismic evidence is an important consideration in the interpretation of any paleoseismic record, the characteristics of the geomorphic and geological setting, and the relative rate of tectonic processes to non-tectonic geomorphic processes, will control whether evidence of past earthquakes is preserved[35]. Ideally, paleoseismic studies consider these factors in site selection and interpretation; while it is acknowledged that missed events are a possibility, it is not clear that this should lead to a systematic bias in our statistics towards or away from periodicity.

### Assessment of prediction error

For each fault, we removed the last event in the record in order to carry out retrospective forecasts using both the model-averaging and single-best model approaches. The single-best model remained the same as that for the full dataset for 49% of the fault segments (46 out of the 93). It appears that the single-best model is more likely to change for fault segments with fewer recorded earthquakes. The number of earthquakes along fault segments for which the single-best model changed is about 27% (95% CI: 11%–39%) fewer than for those fault segments for which the single-best model remained the same. This demonstrates the large model uncertainties for paleoearthquake data, again suggesting that a model-averaging approach is preferable.

Figure 5 shows the 95% credible intervals of the forecast of the last earthquake occurrence time, with 0 representing the mean of the recorded last earthquake occurrence time. Out of the 93 fault segments, the model-averaged forecast successfully covered 79 of the mean true occurrence times. The forecasts from the Poisson process successfully covered 89 of the mean true occurrence times, which at

first glance may suggest that it outperforms the model-averaged forecasts. However, it does this by having much wider credible intervals (on average twice as wide as the credible interval from the model-averaged forecast); i.e. it has much more uncertainty in the forecast. When examined in more detail, we see that model-averaged forecasts routinely outperformed Poisson process forecasts, provided that there were sufficient events left in the record. Specifically, about 83% (77 of the 93 fault segments) of the model-averaged forecasts have much smaller mean squared errors (MSEs) than the forecasts from the Poisson process (Supplementary Fig. 7). MSE is the average squared difference between the forecast value and the true value, which is equal to the sum of the variance and the bias squared, and provides a measure of the trade-off between accuracy and precision. For about half of the fault segments (45 of the 93), the MSEs of the forecasts from the Poisson process are more than twice of that from the model-averaged forecasts (Supplementary Fig. 8). The 14 fault segments for which the model-averaged forecast 95% credible interval didn't cover the true mean were characterised by few events being left in the record: 6 had only 4 events left in the record and thus too few to fit models with more than two parameters, while a further 7 fault segments had fewer than 7 events. Even though in some situations with a small number of events in the record, the less informative, more uncertain Poisson-based forecasts seem to cover the true value, the majority of fault segments with small numbers of events are still better represented by the model-averaging approach (e.g., 23 out of the 30 fault segments which had 4 events in the retrospective forecasts have model-averaged forecasts with much smaller MSE than the Poisson forecasts). It is anticipated that for most hazard modelling purposes the smaller errors associated with the model-averaged forecasts favour their use. Having said this, the Poisson process may still be a valuable model when limited data are available, which is the case for many fault segments that are not included in this study because they have less than five events in the record.

The MSEs of the retrospective forecasts from the single-best model approach are very close to those from the model-averaging approach for the majority of the fault segments (Supplementary Fig. 9), all within two times relative difference. Model-averaging with WAIC weights is not usually designed to achieve a better MSE than the single best-model approach. However, when there is some uncertainty as to the best model, model-averaging outperforms a single best-model primarily in terms of better representing all the uncertainties.

The Bayesian model-averaging approach presented here explicitly considers model uncertainty based on the data and associated measurement errors, rather than relying on selection of a best model. Retrospective testing shows that model-averaging provides more informative and accurate forecasts compared with a single-best model approach or assuming a Poisson process (i.e., random earthquake recurrence). The earthquake probabilities presented in this study also provide a testable hypothesis of future earthquake occurrence that can be evaluated at a global scale.

## Methods
### Models

For each fault segment, we obtain 100 sequences of Monte Carlo (MC) samples for the ages of the sequence of large earthquakes in the paleoseismic record (see Data and Resources section). Each sequence of MC samples is then considered a realisation of the occurrence times of large earthquakes along that fault[9,36]. We denote them by $t_{k0} < t_{k1} < \ldots < t_{kN_i} \leq T$, where $k = 1, 2, \ldots, 100$ denotes the $k$th MC sample, $N_i$ denotes the number of earthquakes in the record for the $i$th fault segment with $i = 1, 2, \ldots, 93$, and $T$ denotes the censoring time which we take as the year 2022. The inter-event times for the $k$th MC sample of the $i$th fault are then $x_{k1} = t_{k1} - t_{k0}, \ldots, x_{kN_i} = t_{kN_i} - t_{k(N_i-1)}$.

For each earthquake record, we fit the following five models. The first is a Poisson process with occurrence rate $Z_k\lambda$ for the $k$th sequence

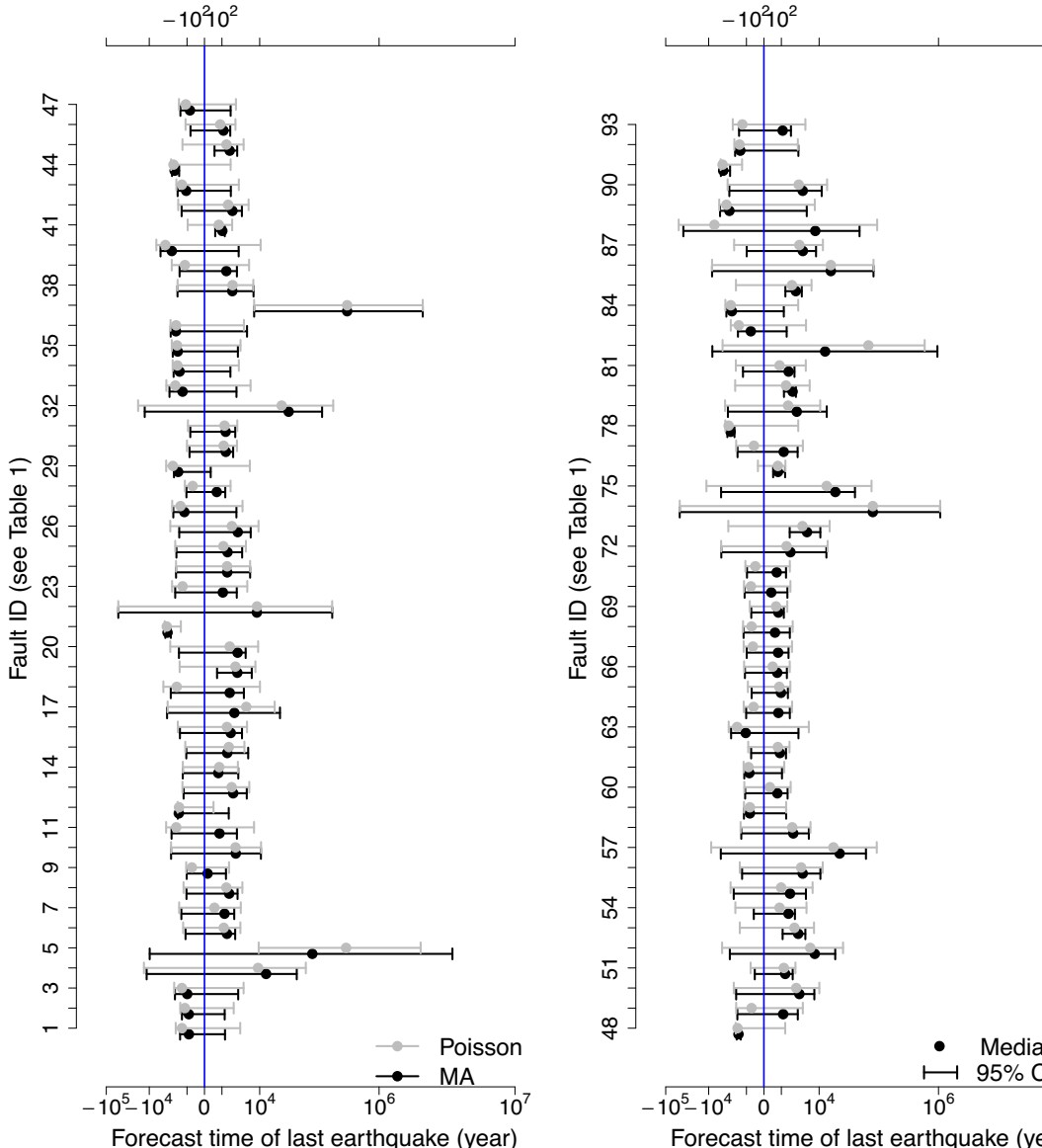

**Fig. 5 | Retrospective forecast of the occurrence time of the last earthquake.** Fault ID is numbered as per the list in Table 1. The forecasts from the model-averaging (MA) approach and the Poisson process are presented here. Markers show median and 95% credible intervals of the forecasts. We centred the estimated values by subtracting the mean occurrence time of the last earthquake in the paleoseismic records. 95% CI 95% credible interval.

of MC samples, where $Z_k$ is a random variable ($k = 1, \cdots, 100$) capturing the similarities between the 100 MC samples for each fault. The second model is a Gamma renewal process with the inter-event times for the $k$th sequence of MC samples following a Gamma distribution with probability density function

$$f(x; \alpha, \lambda, Z_k, Y_k) = \frac{1}{\Gamma(Z_k \alpha)} (Y_k \lambda)^{Z_k \alpha} x^{Z_k \alpha - 1} \exp\{-Y_k \lambda x\} \quad (1)$$

where $Z_k \alpha$ and $Y_k \lambda$ are shape and rate parameters, with $Z_k$ and $Y_k$ being random variables ($k = 1, \cdots, 100$) capturing the similarities between the 100 MC samples for each fault. The third model is a Weibull renewal process with inter-event times for the $k$th sequence of MC samples following a Weibull distribution with probability density function

$$f(x; \alpha, \lambda, Z_k, Y_k) = Z_k \alpha (Y_k \lambda)^{Z_k \alpha} x^{Z_k \alpha - 1} \exp\{-(Y_k \lambda x)^{Z_k \alpha}\} \quad (2)$$

where $Z_k \alpha$ and $Y_k \lambda$ are shape and rate parameters. The fourth model is a Brownian Passage-Time (BPT, also called inverse Gaussian) renewal process with inter-event times for the $k$th sequence of MC samples following a BPT distribution with probability density function

$$f(x; \mu, \beta, Z_k, Y_k) = \sqrt{\frac{Z_k \mu}{2\pi (Y_k \beta)^2 x^3}} \exp\left\{ -\frac{(x - Z_k \mu)^2}{2 Z_k \mu (Y_k \beta)^2 x} \right\} \quad (3)$$

where $Z_k \mu$ and $Y_k \beta$ are the mean and coefficient of variation of the distribution. The fifth model is a lognormal renewal process with inter-event times for the $k$th sequence of MC samples following a lognormal distribution with mean $Z_k + \mu$ and standard deviation $Y_k \sigma$, both on the log scale.

### Estimation
Given the $k$th MC sample of the earthquake occurrence times $t_{k0}, t_{k1}, \ldots, t_{kN_i}$ along the $i$th fault, with final censoring time $T$, the

likelihood of the $k$th MC sample for each model is

$$L(\theta_k; t_{k1}, \ldots, t_{kN_i}, T) = (1 - F(T - t_{kN_i}; \theta_k)) \prod_{j=1}^{N_i} f(t_{kj} - t_{k(j-1)}; \theta_k), \quad (4)$$

where $k = 1, \cdots, 100$, $\theta_k = (\lambda, Z_k)$ for the Poisson process, $\theta_k = (\alpha, \lambda, Z_k, Y_k)$ for the Gamma and Weibull renewal processes, $\theta_k = (\mu, \beta, Z_k, Y_k)$ for the BPT renewal process, and $\theta_k = (\mu, \sigma, Z_k, Y_k)$ for the lognormal renewal process

The different MC samples from the same fault should have similar recurrence patterns. To reflect this, we assume that both $Y_k$ and $Z_k$ follow a distribution with mean 1, i.e.,

$$Y_k \sim Gamma(1/\sigma_Y^2, 1/\sigma_Y^2), \qquad Z_k \sim Gamma(1/\sigma_Z^2, 1/\sigma_Z^2), \quad (5)$$

where $\sigma_Y$ and $\sigma_Z$ are the standard deviations of $Y_k$ and $Z_k$, respectively.

A Markov Chain Monte Carlo (MCMC) algorithm generates samples from the joint posterior distribution of $\theta_k$, $\sigma_Y$ and $\sigma_Z$ given the occurrence times $t_{k0}, t_{k1}, \ldots, t_{kN_i}$ from the $k$th MC sample and the censored time $T$, using software JAGS and the `R2jags` package in R[37]. We use three chains, half-normal priors for $\alpha$ and $\beta$, and weakly informative half-$t$ prior distributions for the variance parameters $\sigma_Y$ and $\sigma_Z$[38], i.e.

$$\alpha \sim N(0, 100^2)T(0,), \qquad \lambda \sim N(0, 100^2)T(0,),$$
$$\mu \sim N(0, 100^2)T(0,), \qquad \beta \sim dt(0, 0.04, 3)T(0,),$$
$$\sigma \sim dt(0, 0.04, 3)T(0,), \qquad \sigma_Y \sim dt(0, 0.04, 3)T(0,),$$
$$\sigma_Z \sim dt(0, 0.04, 3)T(0,).$$

For the MCMC algorithm, we use three chains with 5,010,000 iterations, discarding the first 10,000 iterations as burn-in, and use a thinning rate of 1000. The scale reduction factors of the Gelman-Rubin convergence diagnostic are all less than 1.02, indicating convergence[39,40].

## Model-averaged forecasts

To calculate Bayesian model-averaged forecasts, we combine the posterior distributions of the forecast under each model using model weights. We use prediction-based Bayesian model-averaging (PBMA)[15] with the model weights calculated using the Watanabe-Akaike Information Criterion (WAIC)[20]. This is far less sensitive to the priors for the parameters than classical Bayesian model-averaging (CBMA), which uses posterior model probabilities[15]. PBMA is sometimes referred to as Bayesian model combination[41]. Unlike CBMA, it does not involve the assumption that one of the models is true. The WAIC for model $k$ is calculated as

$$\text{WAIC}_k = -2\sum_{i=1}^{n} \log(p(y_i|y, k)) + 2p_k \quad (6)$$

where $p(y_i|y, k)$ is the pointwise posterior predictive density from model $k$, which can be estimated using the mean of the posterior MCMC sample of $p(y_i|\theta_k, y, k)$ for model $k$; and $p_k$ is a correction for overfitting. A common choice for $p_k$ is

$$p_k = \sum_{i=1}^{n} \text{var}\{\log p(y_i|\theta_k, y, k)\}, \quad (7)$$

where each term in the summation can be estimated by taking the variance of the posterior MCMC sample of $\log p(y_i|\theta_k, y, k)$ for model $k$. The WAIC weight for model $k$ is given by

$$p(k|y) \propto \exp\left[-(\text{WAIC}_k - \min_i \text{WAIC}_i)/2\right]. \quad (8)$$

WAIC is a prediction-based criterion, analogous to AIC in the non-Bayesian setting, and use of Eq. (8) to define the model weights is

motivated by the form of AIC weights[42]. Alternative approaches to prediction-based model averaging in seismology have been proposed[43,44]. We prefer to make use of WAIC weights for the following reasons. Model selection using WAIC has the desirable property, in large samples, of being equivalent to Bayesian leave-one-out cross-validation (B-LOO)[45]. When B-LOO is used in model averaging it is known as Bayesian stacking, and has the useful property that, for large samples, it leads to the best linear combination of the posterior distributions of the forecasts from each model, whereas CBMA will lead to use of the posterior distribution of the forecast from the single best model[41–43] (which is why some authors refer to CBMA as a tool for model selection[41]). We would therefore expect WAIC weights to provide a close-to-optimal linear combination of the posterior distributions of the forecasts from each model, whilst being much less computationally-intensive than Bayesian stacking.

For each model, we can obtain a posterior MCMC sample of the forecast quantity: either the forecast occurrence time of the next large earthquake or the forecast probability of at least one large earthquake occurring in the next 50 years along the specified fault segment. These forecasts are conditioned on the fact that there was no large earthquake between the last large earthquake occurrence time and the year 2022. The five posterior MCMC samples of the forecast quantity were combined into one model-averaged posterior sample by randomly taking the value from one of the five posterior samples at each iteration. The probability weights for the random sampling are the WAIC weights for the corresponding five models.

## Data availability

All data used in this study have been deposited in Zenodo https://zenodo.org/records/4131308[46].

Long-term earthquake records were compiled from previously published studies for 93 individual fault segments globally. The majority of the data were taken from a previous compilation[9], with additional records (mainly from China[47]) added to extend this database. The data primarily consists of results from paleoearthquake studies from a single site on a fault segment, supplemented by historical data where they exist. We select records that contain at least five events. Some studies have considered evidence from more than one site on a particular fault segment; in this case, we use an earthquake record that combines data from the different sites only if provided in previously published studies, and do not attempt to combine earthquake records ourselves. For subduction zones, earthquake records are necessarily derived from proxy evidence (e.g. paleogeodesy, paleotsunami or turbidite studies), rather than direct on-fault evidence, and again we rely on the assessment of the relevant study authors for attribution of the evidence to the fault in question. For some faults in our database (in particular the San Andreas Fault) many studies have been carried out on neighbouring fault segments, some of which are known to have ruptured together in past historical earthquakes. For this study we treat each record independently, and do not attempt to correlate past ruptures or consider the probability of co-rupture of multiple segments and/or faults in our model, as has been done in other studies focused on data-rich regions[24,48].

## Code availability

All R code for the analysis and plots is available in Zenodo https://doi.org/10.5281/zenodo.10511930[49]. Python code for the Monte Carlo sampling of the earthquake chronologies is available in Zenodo https://zenodo.org/records/4131308[46].

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

## Acknowledgements

T.W. and M.S. are partially funded by MBIE Endeavour Fund (Smart Ideas, Grant number UOOX2206). J.G. publishes with the permission of the CEO, Geoscience Australia. We thank Hua Pan for help with understanding the catalogues from China. We thank Stephen Read for help plotting Fig. 2 and Gareth Davies for constructive comments. We thank the New Zealand eScience Infrastructure (NeSI) for providing access to the supercomputers.

## Author contributions

T. W. conceptualised the study, designed and performed all data analyses. T. W., J. G., and M. B. wrote the manuscript, which was then revised by all the authors. J. G. compiled the data and generated the Monte Carlo samples of the earthquake chronologies. T. W. and M. B. designed the illustrations. D. F. provided the R code for model-averaged forecasts, helped with some of the data analysis. J. Z. provided advice on model-averaging and participated in the initial simulation study. M. S. provided advice on the paleoseismic data and earthquake hazard forecasting. P. D. helped with model design and some of the data analysis. J. K. participated in the initial simulation study.

## Competing interests

The authors declare no competing interests.
