## [Peer Review File · Nature Communications]

Earthquake forecasting from paleoseismic recordsEditorial Note: Parts of this Peer Review File have been redacted as indicated to remove third-party material where no permission to publish could be obtained.

REVIEWER COMMENTS

Reviewer #1 (Remarks to the Author):

The authors have proposed a Bayesian model-averaging approach for forecasting large earthquakes along active faults. They emphasise the importance of model uncertainty in the conventional single-model approaches and ensemble all the candidate models with their weights derived from the model selection criterion WAIC to mitigate the effect of model uncertainty on forecasts.

Selection of recurrent-time distribution has long been controversial for long-term earthquake forecasts. The authors argue that no single model consistently provides the best fit across the global catalogs, or even within the individual fault systems. Their approach presents a plausible solution in this situation. Additionally, the authors discussed the periodicity of earthquake recurrence and revealed through regression analysis that the degree of periodicity depends not only on the occurrence rate (as discussed in the previous studies) but also on the faulting type and the tectonic region.

The authors provide the probability forecasts in the next 50 years for the global 93 fault segments. Their retrospective analysis shows good accuracy of the proposed approach for most of the fault segments. The paper is well organised and insightful. However, I'd like to provide feedback on several specific points:

L86: The authors describe that the AIC can't provide guidance as to how much better a particular model is compared to other models. However, since the AIC is an estimator of the prediction error of a fitted model in terms of the log-likelihood of a new observation, it can indeed assess the relative predictive competence of different candidate models.

L148: Though the authors have revealed relationships between the number of events and the single-best model in their catalogs, they do not provide discussion on characteristic behavior of earthquake recurrence in each single model. For example, calculating the normalised recurrence intervals, which are divided by the mean recurrence intervals of the respective fault segments, grouping them by the selected single-best models, and comparing the histograms between the single-best models might help us to observe some characteristic recurrence behavior in each model. In this case, I suggest referring the following paper:

Nishenko, S. P., & Buland, R. (1987). A generic recurrence interval distribution for earthquake forecasting. *Bulletin of the Seismological Society of America*, 77(4), 1382-1399.

L207-222: The ranges of the CIs in this paragraph appear too short compared to the sample size of 93 and the dispersion of the data points in Figure 4. I suspect that this might be because all the 100 MC samples and/or 5000 MCMC samples for each segment are used as the data points in evaluating the CIs. If this is the case, it is not appropriate to discuss the statistical significance using the CIs from these leveraged samples.

L244: From Figure S4, the authors have shown that the forecast values from the model-averaging approach have much smaller mean square errors (MSEs) than those from the Poisson process. However, it is difficult to see the relative improvement in accuracy from this figure because the fault segments have a wide range of scales in their recurrence times. Therefore, I suggest adding a figure and comments on the relative differences in MSEs between the model-averaging approach and the Poisson process.

L252: favours -> favour ?

L265: The accuracy of forecasts for the single-best model approach does not seem to be discussed in the retrospective tests in Section 5.

L292: The sentence ends with '.'.

Reviewer #2 (Remarks to the Author):

The paper aims at introducing a new methodology to estimate large earthquake occurrence on faults. Although the topic is of great interest and the paper contains a very interesting approach, I have some concerns that should be carefully addressed; some of them might be critical.

Below I list my main concerns

- This is likely the most critical point. In the "Introduction" section, the authors did not quote recent bibliography on the earthquake recurrence models on faults, such as the recent Neely et al (SRL, 2023) and the work made in UCERF3 (Field et al., 2015: Long-Term Time-Dependent Probabilities for the Third Uniform California Earthquake Rupture Forecast(UCERF3), BSSA). Interestingly, Field et al (2015) discuss the impact on the earthquake occurrence models (in this case, the Brownian Passage Time also used here) when some assumptions are relaxed, such as relaxing a strict faults segmentation. In this paper, the authors are assuming a strict faults segmentation which is, at best, debatable (so many recent examples, including Kaikoura eqk). If this assumption is relaxed, the models used by the authors may be incoherent and/or inconsistent. What is the impact of relaxing faults segmentation, and what is the reliability of such an assumption? These are important issues that should be carefully and satisfactorily addressed in the paper.

- I am not a paleoseismologist, but my feeling is that the data that can be collected from a fault may be filtered. In essence, if two large earthquakes occur nearby in time, they cannot be distinguished (in many cases). This means that the observations have a low-frequency filter, and removing short inter-event times the data appear more periodic than what they are. Maybe a short discussion on the time resolution of the catalogs may be worthwhile.

- Although I am very positive on the "averaging" approach proposed by the authors I think that the authors should modify their terminology and their procedure to assign weights. The Bayesian Model

Averaging (BMA) is something different from what is made by the authors, but, more important, BMA is not an averaging tool, but a model selector. This problem has been deeply discussed by Monteith et al. (2011; Monteith, K., J. L. Carroll, K. Seppi, and T. Martinez (2011). Turning Bayesian model averaging into Bayesian model combination, in Proc. of International Joint Conference on Neural Networks, San Jose, California, 31 July–5 August 2011, 2657–2663.). This problem has been found and discussed also in seismology (Marzocchi Zechar and Jordan, 2012, BSSA). The way in which the authors calibrate the weights, using a quantity that is proportional to probability (see their equation 8), is similar to BMA and it might have the same problems (see the literature mentioned above). It might be interesting for the authors to know that more recently, Herrmann and Marzocchi (GJI, 2023) discussed the optimal weighting scheme for an averaging procedure, which is based on a fundamentally different logic. Here I suggest a more careful re-thinking about the weighting scheme and a solid justification of the method used. Maybe just using a quantity proportional to WAIC instead of using the exponential, it might improve the problem (This is just a suggestion).

- Again, despite being positive about the method, I am wondering what is the reason for what faults behave differently (they follow different distributions); is there a different physics? Or what else? In my understanding, the Weibull distribution is the best performing on faults which have a longer dataset. I am wondering what happens if we use the Weibull model for all faults. Of course, in some cases, the Weibull model will perform less well than other models, including the average model, but what is the information gain per event? One model can perform worse than another for a small difference. In this case, the differences do not have any practical importance. I suggest the use of some metric, like the information gain per event, to show how much a model performs better than others and discussing the results.

- on a minor side, LINE 30 in the Abstract. I never liked too much the word "robust", because it is unclear what it means. At least to me.

Reviewer #3 (Remarks to the Author):

The manuscript by Wang et al presents a Bayesian statistical analysis of a large number of paleoseismic data sets with the apparent goal of developing an improved forecasting method. The central claims of this manuscript, as I see them, are as follows: 1) Bayesian analysis allows uncertainties in earthquake-recurrence model selection to be more fully accounted for in subsequent forecasting; 2) No single model can account for all of the observed paleoseismic sequences, even those from faults with similar kinematics, strain rates, or tectonic settings; 3) Retrospective forecasting shows that a “model averaging” approach to estimating earthquake occurrence is more informative and accurate than any single other model, including the Poisson (i.e. random earthquakes) model; 4) Various segments of the San Andreas fault system have a high likelihood of rupture (i.e. up to 70% chance) over the course of the next 50 years; and 5) the large dataset analyzed here allows a variety of generalizations about the nature of

earthquake recurrence as a function of strain rate, kinematics, and/or tectonic setting.

Of these claims, it is my opinion that (1) and (2) are generally well supported by the data and associated arguments. Claim (3), however, is a much more complicated question, and to be honest, I feel that the manuscript on some level over intellectualizes it. There are practical/societal matters that need to be more fully considered (discussed below). Claim (4) is also well supported by the data and associated discussion, but I do not see where it departs appreciably from a variety of other active forecasting methods that have been applied to the San Andreas system. This kind of begs a question (and I do mean this respectfully) of what exactly is gained by using this new approach when compared with other methods? Finally, claim (5) is also generally well supported here, although at least some of the subclaims related to it have been proposed by other authors who need to be cited. Considering aspects of the presentation, the writing is clear and concise. The quality and utility of the illustrations is similarly top notch. My recommendation is that this manuscript be further considered for publication following moderate revisions.

I could probably summarize many of my concerns with this manuscript using a single question that I would ask the authors to consider: Why is this approach fundamentally and necessarily better than what has been done previously? In this way, we need to realize that earthquake forecasting is not just an interesting scientific exercise. Rather, it has explicit practical goals, which vary depending on who is asking the question. As a society we make important decisions about how and where to distribute finite resources based on such forecasts. It may be most "correct" for a forecasting approach to more fully consider uncertainties in recurrence-model selection, but that only really matters if doing so leads to a demonstrably more accurate forecast. This is the central issue with all earthquake forecasting work - it's going to be a very long time before we really KNOW if these approaches are of any real utility or which are better than others. Retrospective forecasting is certainly one way to get a tiny glimpse at the answer. It's often unsatisfying though, and it is for me in this case as well, in large part because the confidence intervals for the next event are so large (an unfortunate reality of uncertain datasets, as this manuscript often points out). If our confidence interval for the occurrence of the most recent event in the data set spans hundreds or thousands of years, and we apply it to a record with a mean recurrence interval of about 100 years, is it really surprising/revealing that we successfully capture the event within the predicted interval? So respectfully, why is the approach presented here preferred? How would we pitch it to an insurer? A local government? A builders association? The factor that all of these entities are interested in is less a matter of the binary occurrence of an earthquake within some window of time, but rather the per-year likelihood of a catastrophic event (often compared with the anticipated useful lifetime of a structure). By decreasing the confidence window of occurrence, the method explicitly raises the per year likelihood of an event, and that has implications. I understand this is a manuscript that is focused on science, so maybe these questions do not need to be explicitly answered, but I do think that the subject matter really does demand at least some engagement with the practical matters at hand.

Toward the point above, it also seems that any manuscript presenting a new approach to earthquake forecasting should provide some detailed comparisons with previous forecasting work. For example, the earthquake occurrence probabilities for various segments of the San Andreas presented here strike me as largely the same as those presented by the Uniform California Earthquake Rupture Forecast. UCERF also includes a time-dependent forecast in addition to a Poisson. I'll grant that the approach here

presents a less uncertain forecast than a Poisson alternative whilst also being broadly as accurate (per the retrospective forecast), but what about other time-dependent models such as UCERF? Again respectfully, why is the approach presented here better (this is not to express doubt that it may be, but rather that it is not clear WHY it necessarily is)?

Finally, the manuscript consistently does a bit of a hit job on the use of the Poisson model in earthquake forecasting. I think that should be dialed back a bit. Yes, I too think the idea that earthquakes would occur randomly in time is a bit rubbish from a mechanics perspective, and it is not supported by the current state of the paleoseismic literature as the authors seem to agree. That being said, there are practical reasons for the adoption of a Poisson model within earthquake forecasting as described in the line referenced comments (i.e. it is a decent choice when the date of the most recent event is not known).

In summary, this is interesting and high quality work. Having communicated with some of the authors on the subject of earthquake statistics and forecasting previously, I know that their desire to do good science is genuine. I think this work too will soon be worthy of publication, but given its nature, some considerably more nuanced discussion of why this approach is better than the alternatives is required.

Randy Williams
22 August 2023

Line Referenced Comments

18-20: Interesting question really - does better incorporation of uncertainty lead to a more robust forecast? Depends on what a "robust" forecast is. Seems at least possible that we end up with a forecast that is no more or less accurate than was the case prior to using this Bayesian approach.

21-22: Again, this is cool, but minimizing the forecast uncertainty is only a useful thing if the forecast itself is accurate. The central dilemma of earthquake forecasting! We'll need decades (if not centuries) to find out.

40-47: Could simplify this section for the novice with some more plaintext terminology. The Poisson process describes events that effectively occur randomly in time. And the antithesis of that for periodic sequences of course.

57-60: Maybe rephrase to better emphasize the severity of the problem. I'd wager that relatively few of those 90 records actually have anywhere near 20 distinct events. In my experience records with more than 10 events are quite rare. Many have fewer than 5.

60-61: It might, certainly, but does it? Again, we really don't know at this point.

77: First occurrence of (what I assume is) Brownian passage time. Please define.

98-99: See comment above about record length.

108: Perhaps this comment is too “in the weeds” but I feel like we have known for a while that a lognormal distribution of interevent times is not an appropriate model for earthquakes, largely because its hazard function actually decreases through time from some maximum value.

147-150: Quite interesting though.

169: Could just call this “27” time as likely?

207-222: Some of this discussion makes one speculate as to whether variations in strain rate or kinematics have an appreciable impact on record length and/or event preservation. Maybe worth a teaser sentence?

211-215: Pretty sure that Berryman made a similar argument in the 2011 Alpine fault paleoseismology paper. Should probably give them a citation.

251-252: I think this statement needs a bit of justification. Why is having smaller errors necessarily preferable from a hazards forecasting perspective? As stated above, it’s only really a good thing if the forecast is accurate.

253: Although I am also in favor of time-dependent approaches, we can probably cut the Poisson model a little bit of slack here. It does have its uses from a hazards perspective, particularly if we have some sense of the average interseismic period length but not a date on the most recent slip event. We see in the retrospective forecast that in a select few cases smaller errors caused the analysis to miss the most recent event.

235-240: Yes, the Poisson model does yield larger uncertainties and thus more frequent “overlap” with the actual events, but to be fair, the model average approach also yields fairly large uncertainties in an absolute sense. In the example of the San Andreas, if the forecast produces a 95% confidence interval for the next event that is a period of several hundred years, whilst the actual data suggest that a typical recurrence interval is on the order of about 100 years, it is a bit unsurprising that the forecast would capture the event within its uncertainty. Respectfully, I feel that I have to ask what exactly we learn from this? Certainly it could be interpreted to show that your approach isn’t fundamentally off base, but why is this necessarily better than all of the other hazards forecasts that are available for the San Andreas (or other fault systems)?

Also, the bit about the model averaging approach “outperforming” the Poisson is a bit complicated as written. You state that the model averaging approach successfully covered 79 of the true occurrence times, but then state that the model averaging approach outperforms Poisson because 81 of the model averaging results had lower mean squared errors. Does that not then by definition include 2 records for which your retrospective forecast failed (i.e. the CI did not include the true occurrence time)? Strange to count those as successes of the approach.

Finally, please consider comments above regarding the utility of the Poisson process in cases where the date of the most recent event is unknown. There is practical utility in the time-independent approach in

some cases, even if we believe that fault behavior is, in fact, time dependent.

251-252: Again, this is a really tricky area of application. Please justify why this is favored, remembering WHY we want to forecast earthquakes accurately in the first place. We could of course be conservative/cautious and assume that any place that has a large fault that moved within the last 50 ka has a very high earthquake risk, but that would not do good things for local economies / insurance / building budgets. Although I do like this manuscript, I feel it has a tendency to treat seismic forecasting as little more than an intellectual exercise by focusing on what is most “correct” about fault behavior and uncertainty minimization. Neither of these factors are readily translated into accuracy/reliability/usefulness of the forecast, although they may be components of them. In a practical sense, its less about correctly predicting an earthquake within some interval of time. Rather, it's about calculating a per year likelihood of a catastrophic event for a particular area. Your approach by definition maximizes that likelihood by minimizing the predicted interval.

265-266: I did not see a comparison between the model averaging approach and the “single best model” approach in the retrospective forecasting data?

267-269: This might be a stronger point to the work that is not being fully appreciated by the line in question. If we are ever to validate whether our hazards forecasts are meaningful (in the absence of waiting a few hundred years to find out), it may be best achieved by looking at the global distribution of forecasts and how often they ring true.

RESPONSES TO REVIEWER COMMENTS

Reviewer #1 (Remarks to the Author):

The authors have proposed a Bayesian model-averaging approach for forecasting large earthquakes along active faults. They emphasise the importance of model uncertainty in the conventional single-model approaches and ensemble all the candidate models with their weights derived from the model selection criterion WAIC to mitigate the effect of model uncertainty on forecasts.

Selection of recurrent-time distribution has long been controversial for long-term earthquake forecasts. The authors argue that no single model consistently provides the best fit across the global catalogs, or even within the individual fault systems. Their approach presents a plausible solution in this situation. Additionally, the authors discussed the periodicity of earthquake recurrence and revealed through regression analysis that the degree of periodicity depends not only on the occurrence rate (as discussed in the previous studies) but also on the faulting type and the tectonic region. The authors provide the probability forecasts in the next 50 years for the global 93 fault segments. Their retrospective analysis shows good accuracy of the proposed approach for most of the fault segments.

The paper is well organised and insightful. However, I'd like to provide feedback on several specific points:

L86: The authors describe that the AIC can't provide guidance as to how much better a particular model is compared to other models. However, since the AIC is an estimator of the prediction error of a fitted model in terms of the log-likelihood of a new observation, it can indeed assess the relative predictive competence of different candidate models.

Reply: We agree that this statement is misleading, and we have therefore removed it.

L148: Though the authors have revealed relationships between the number of events and the single-best model in their catalogs, they do not provide discussion on characteristic behavior of earthquake recurrence in each single model. For example, calculating the normalised recurrence intervals, which are divided by the mean recurrence intervals of the respective fault segments, grouping them by the selected single-best models, and comparing the histograms between the single-best models might help us to observe some characteristic recurrence behavior in each model. In this case, I suggest referring the following paper:

Nishenko, S. P., & Buland, R. (1987). A generic recurrence interval distribution for earthquake forecasting. *Bulletin of the Seismological Society of America*, 77(4), 1382-1399.

Reply: We thank the reviewer for this suggestion. We carried out this analysis as suggested by the reviewer. For each fault segment, we calculated the normalised recurrence intervals for each MC sample, and then reported the median and the 2.5% and 97.5% quantiles of the standard deviations of the normalised recurrence intervals calculated for the 100 MC samples. However, we didn't find any characteristic recurrence behaviour for the fault segments that have the same single-best model. See the figure below. We did observe that fault segments with higher rate of earthquake occurrences appear to have smaller standard deviations of the normalise

recurrence intervals. We have added this discussion in the revised manuscript, Lines 196-203, which is also included below.

“Previous studies (Sykes and Nishenko, 1984; Nishenko, 1985; Nishenko and Buland, 1987) have shown that the standard deviations of the scaled (divided by the mean value) inter-event times along several fault segments appear to be constant. For each of the 93 fault segments, we calculated the scaled inter-event times for each MC sample, and then reported the median and the 2.5% and 97.5% quantiles of the standard deviations of the scaled inter-event times calculated for the 100 MC samples. There does not seem to be any characteristic recurrence behaviour for the fault segments that have the same single-best model. Fault segments with higher rate of earthquake occurrences appear to have smaller standard deviations of the scaled inter-event times (Supplementary Fig. 4).”

L207-222: The ranges of the CIs in this paragraph appear too short compared to the sample size of 93 and the dispersion of the data points in Figure 4. I suspect that this might be because all the 100 MC samples and/or 5000 MCMC samples for each segment are used as the data points in evaluating the CIs. If this is the case, it is not appropriate to discuss the statistical significance using the CIs from these leveraged samples.

Reply: We are grateful for the reviewer's comment related to this part of the analysis. This made us realise that we had not allowed for both sources of errors in the regression model: process error (which would be present even if we knew the true values of the shape parameter) and sampling error, i.e. the uncertainty in our estimation of the true value of the shape parameter. In the revised manuscript, we added both sources of errors in the regression model and have updated the CIs which are wider than in the original manuscript. See Lines 273-296 in the revised manuscript, which is also included below. R code related to all the data analysis is provided in the supplement.

“A regression model for the relationship between the shape parameter of a Weibull renewal process and the earthquake rate, tectonic setting, faulting type, and the number of earthquakes of each of the 93 fault segments suggests that when the earthquake rate increases 10 fold (e.g., from 0.0001 to 0.001), the shape parameter of the Weibull renewal process increases by about 24% (95% CI: 5%-45%) (Figure 4). This suggests that fault segments with higher earthquake rates tend to have more periodic behavior. The shape parameter of the Weibull renewal process for fault segments located in an intraplate setting is about 87% (95% CI: 51%-148%) of that for fault segments located at or near plate boundary. Although this the 95% CI is wide and covers 100%, the posterior density plot (Supplementary Fig. 5) suggests that there is a high probability (0.7) that the latter may be more periodic than the former. There are only five fault segments from an intraplate setting, so to reach a more robust conclusion, more data from intraplate fault segments are needed. The shape parameter of the Weibull renewal process for fault segments located in an active intraplate setting (predominantly faults in central China) is about 1.4 (95% CI: 1.1-1.9) times of that for fault segments located at or near plate boundary, suggesting that the former appear to be more periodic than the latter. The shape parameter of the Weibull renewal process for fault segments located in a subduction region is about 1.9 (95% CI: 1.1-3.3) times of that for fault segments located at or near plate boundary, suggesting that the former appear to be more periodic than the latter. On average, the Weibull shape parameter for reverse faults is about 69% (95% CI: 46%-104%) of that for normal faults, suggesting that large earthquakes recur more periodically on normal faults than on reverse faults.”

L244: From Figure S4, the authors have shown that the forecast values from the model-averaging approach have much smaller mean square errors (MSEs) than those from the Poisson process. However, it is difficult to see the relative improvement in accuracy from this figure because the fault segments have a wide range of scales in their recurrence times. Therefore, I suggest adding a figure and comments on the relative differences in MSEs between the model-averaging approach and the Poisson process.

Reply: We have included Supplementary Fig. 7 in the revised supplement to show the relative differences in MSEs between the Poisson process and the model-averaging approach. We have also added the following comments in the revised manuscript, Lines 319-321.

“For about half of the fault segments (45 of the 93), the MSEs of the forecasts from the Poisson process are more than twice of that from the model-averaged forecasts (Supplementary Fig. 7).”

L252: favours -> favour ?

Reply: Thank you. We have changed this.

L265: The accuracy of forecasts for the single-best model approach does not seem to be discussed in the retrospective tests in Section 5.

Reply: Thank you. We have added Supplementary Fig. 8 in the revised supplement to show the relative differences in MSEs between the single best model and the model-averaging approach, and added this discussion in the revised manuscript. See Lines 336-341, which is also included below.

“The MSEs of the retrospective forecasts from the single-best model approach are very close to those from the model-averaging approach for the majority of the fault segments (Supplementary Fig. 8), all within two times relative difference. Model-averaging with WAIC weights is not usually designed to achieve a better MSE than the single best-model approach. However, when there is some uncertainty as to the best model, model-averaging outperforms a single best-model primarily in terms of better representing all the uncertainties.”

L292: The sentence ends with ‘..’.

Reply: Thank you. We have fixed this.

Reviewer #2 (Remarks to the Author):

The paper aims at introducing a new methodology to estimate large earthquake occurrence on faults. Although the topic is of great interest and the paper contains a very interesting approach, I have some concerns that should be carefully addressed; some of them might be critical.

Below I list my main concerns.

- This is likely the most critical point. In the "Introduction" section, the authors did not quote recent bibliography on the earthquake recurrence models on faults, such as the recent Neely et al (SRL, 2023) and the work made in UCERF3 (Field et al., 2015: Long-Term Time-Dependent Probabilities for the Third Uniform California Earthquake Rupture Forecast(UCERF3), BSSA). Interestingly, Field et al (2015) discuss the impact on the earthquake occurrence models (in this case, the Brownian Passage Time also used here) when some assumptions are relaxed, such as relaxing a strict faults segmentation. In this paper, the authors are assuming a strict faults segmentation which is, at best, debatable (so many recent examples, including Kaikoura eqk). If this assumption is relaxed, the models used by the authors may be incoherent and/or inconsistent. What is the impact of relaxing faults segmentation, and what is the reliability of such an assumption? These are important issues that should be carefully and satisfactorily addressed in the paper.

Reply: We thank the reviewer for the suggestion of the two references and for giving us the opportunity to clarify the aim of our research study. We have added these two references in the introduction and added discussions around the reviewer's comments. Please see Lines 57-64, 88-99, 171-182, 216-218 in the revised manuscript. We also provide some further discussions below.

We address each of the two studies mentioned by the reviewer individually.

1. The Long-Term Fault Memory Model proposed by Neely et al. (2023) assumes that partial strain release during earthquakes can lead to accumulated strain on a fault, and that therefore the timing of future events is dependent not only on the time elapsed since the most recent event (as in renewal models) but also previous inter-event times, which may or may not lead to an accumulation of strain that brings forward the timing of the next rupture. This model presents an interesting hypothesis, and the authors applied the model to the Mojave section of the San Andreas fault in California. However, Griffin et al. (2020) used the 'memory coefficient' of Goh and Barabasi (2008) to characterise the autocorrelation of successive inter-event times for 80 long-term earthquake records. For the vast majority of earthquake records there was no significant correlation or anti-correlation between the length of successive inter-event times. In addition, Kempf and Moernaut (2021) generated synthetic earthquake records using Monte-Carlo sampling from typically parameterised renewal models and showed that despite these being independent events it is possible to generate records that show positive or negative memory coefficients. These authors argued therefore that caution should be practised with respect to interpreting memory from short paleoseismic records. We therefore did not consider the Long-Term Fault Memory model as a generic model for global large earthquake recurrence, although we do not discount the applicability of the model in some particular situations.

2. The study by Field et al. (2015), of which some aspects relevant to this study are detailed in Field (2015), discusses the use of renewal models in the context of development of unsegmented fault and rupture models in the UCERF-3 model, which is developed for California earthquake rupture forecast.

UCERF-3 consists of four components, namely the fault model, the deformation model, the earthquake rate model, and underlying these three, the earthquake probability model. In their earthquake probability model, they relaxed the strict fault segmentation assumption in UCERF-2 by subdividing the main fault sections into many fine subsections and considering interactions between different subsections. For each individual fault subsection, although interactions between fault subsections are included, the fundamental elastic-rebound theory part of the model is a BPT renewal model.

We aim to use a global record to find a better approach for the elastic-rebound theory part of the earthquake probability model component, that is, a better way to model the inter-arrival times which have been frequently modelled using either the BPT renewal process, because of its physical explanation of the earthquake process (Matthews et al., 2002; Nomura et al., 2011; Field et al., 2015), or the best model among a few candidate renewal processes, based on an information criterion (Ogata, 1999; Rhoades et al., 2011). Once this layer of modelling is built, one can then add interactions between small subsections to forecast large earthquakes along specific individual faults. Improving the forecast from this layer will result in a holistic hazard model that includes fault models, deformation models, earthquake rate models, and earthquake probability models. We use the global data set to demonstrate that there is no universal model and the BPT renewal process is not necessarily the best model. Choosing a different renewal model or choosing to average over several renewal processes may produce a better forecast.

The fundamental question is around fault segmentation, and in particular for the San Andreas which a) has many paleoseismic sites along different fault segments; and b) has been the subject of several other studies to estimate time-dependent earthquake probabilities.

It is also noted that our study attempts to present a globally applicable method, and that in some more well-studied regions such as the San Andreas Fault System it may be appropriate to develop fault-specific approaches such as that presented by Field et al (2015) that build in additional information about fault segmentation and geometry. It is not our intention to repeat such studies. That said, the question of segmentation and co-rupture of multiple fault segments in less frequent, larger magnitude events vs rupture of individual segments in more frequent, smaller magnitude events is significant from a seismic hazard point of view. The challenge arises because, apart from a small number of historical events where we have information regarding whether co-rupture of neighbouring fault segments occurred, uncertainties in the paleoseismic record cannot resolve whether two fault segments ruptured in the one event, or in two separate events, even when dating uncertainties overlap. That is, two separate events a day apart on neighbouring fault segments will look very different to a single event rupturing both

segments from a hazard point of view, yet these two possibilities will be indistinguishable by comparison of paleoseismic records from the two segments. While many studies assume overlapping dating uncertainties to indicate co-rupture (e.g. the ‘stringing pearls’ approach; Biasi and Weldon 2009), these are subject to a number of subjective decisions. For example, in Biasi and Weldon 2009 (p478) overlapping ruptures are arbitrarily discarded based on a series of rules, such as if the median is more than 40 years from the mean, or the width of the combined age distribution is greater than 90 years, and several other criteria. While these all may or may not be considered reasonable criteria to impose, we cannot independently determine from the data alone whether events at adjacent paleoseismic sites that have overlapping age distributions are the same event or different events. At best, paleoseismic records can exclude co-rupture in the case that there is negligible overlap in age uncertainties between sites. Some inference on the likelihood of co-rupture can be gained from analysis of historical surface rupturing events (e.g. Biasi and Wesnousky 2021; Walsh et al 2023), however the problem of combining data from multiple paleoseismic sites is still largely intractable without recourse to some kind of expert judgement.

In the Field et al (2015) approach, time-dependent probabilities are assigned to different ruptures (i.e. different combinations of fault segments) by taking averages of the relevant parameters (mean inter-event time and time of most recent event) over the participating fault segments. To the best of our knowledge, the relative likelihood of two adjacent segments rupturing together in one event vs independently in two events is not constrained by the paleoseismic data. Note that the value for the aperiodicity is assumed (P552 of Field, 2015), while the paleoseismic data is used only to constrain the mean recurrence intervals. The model also ‘assumes the availability of a long-term earthquake-rate model, which gives the frequency of all possible superseismogenic ruptures on a fault or fault system’ (Field et al., 2015 p515). This seems to be quite different to what we are doing here, where we are fitting renewal models to the paleo data directly at each site. The method developed in our study could be used as an input to a UCERF3 style model, in terms of providing more accurate constraints on the mean recurrence intervals and aperiodicity, but it is beyond the scope of this study to implement a UCERF3 approach.

- I am not a paleoseismologist, but my feeling is that the data that can be collected from a fault may be filtered. In essence, if two large earthquakes occur nearby in time, they cannot be distinguished (in many cases). This means that the observations have a low-frequency filter, and removing short inter-event times the data appear more periodic than what they are. Maybe a short discussion on the time resolution of the catalogs may be worthwhile.

Reply: Thank you for raising this issue. Please see our detailed response below. We have included a short discussion in the revised manuscript on the time resolution of the catalogs as suggested by the reviewer. See Lines 285-295.

1. Perhaps the most commonly used event marker in paleoseismic studies is colluvium formed due to collapse of a fault scarp. Where there is a component of dip-slip, fault scarps are often unstable, and if collapse does not occur co-seismically (as is often the case), it is typically seen within days to months, and occasionally a few years after an

earthquake (e.g. many of the 2016 Kaikoura earthquake scarps degraded rapidly, e.g. Kearse et al., 2018). Strike-slip earthquakes often lead to vertical cracking at the surface of the fault plane, and again these are unstable structures that typically collapse and/or infill quickly. Future events that re-rupture the same fault segment will typically fault the deposited colluvium, before deposition of a new (unfaulted) colluvial unit above the previous unit. In this situation, exposure of the faulted and unfaulted colluvial units is an indicator of two separate events, and identification of multiple events is dependent entirely on the stratigraphic and structural relationships of the units, and not the length of time between their formation. i.e., the stratigraphy will look approximately the same regardless of the inter-event time, barring perhaps some inter-event soil formation or other deposition, depending on the environment, in the case of a long inter-event time. Note that the identification of the two events based on stratigraphic relationships is a separate question as to whether current dating techniques can resolve their absolute timing. While preservation of paleoseismic evidence is an important consideration in the interpretation of any paleoseismic record, the characteristics of the geomorphic and geological setting, and the relative rate of tectonic processes to non-tectonic geomorphic processes, will control whether evidence of past earthquakes is preserved (McCalpin and Nelson 2009). Ideally, paleoseismic studies consider these factors in site selection and interpretation; while it is acknowledged that missed events are a possibility, it is not clear that this should lead to a systematic bias in our statistics towards or away from periodicity.

2. As discussed by Williams (2022), if fault re-rupture commonly occurred shortly after previous earthquakes, then we would expect to see this frequently in the historical record of surface rupturing earthquakes, which we largely do not. To quote from that paper:

“It seems important to recognize, however, that such short-term re-rupturing of individual faults or fault sections is almost entirely absent from the historical and modern records of large earthquakes. In California, for example, even the most famous examples of short-term re-rupture of the same fault section had IETs of ~40 yr (Imperial fault, 1940–1979; southern San Andreas Mojave section, 1812–1857; northern San Andreas Santa Cruz Mountain section, 1838–1890–1906). It is important to note that in the case of the San Andreas Mojave and Santa Cruz Mountain sections, these re-rupture events were in fact captured in the paleoseismic records of the affected area (Weldon et al., 2004; Streig et al., 2014). Were short term re-ruptures as common globally as predicted by truly random recurrence, it seems reasonable that at least several examples would have been observed, particularly given the preponderance of active seismogenic faults in tectonically active areas around the world and the length of the historic record in some regions (i.e., thousands of years in Turkey, Jordan, China, and Japan).”

This suggests that it is unlikely that we are systematically missing very short inter-event times and hence biasing our analysis. Note that this is a different issue to the question of correlating events between paleoseismic study sites and resolving whether events

with overlapping age uncertainties at multiple sites are the same event or different events that occurred closely together in time.

Berryman, K.R., Cochran, U.A., Clark, K.J., Biasi, G.P., Langridge, R.M. and Villamor, P., 2012. Major earthquakes occur regularly on an isolated plate boundary fault. *Science*, 336(6089), pp.1690-1693.

Biasi, G.P. and Weldon, R.J., 2009. San Andreas fault rupture scenarios from multiple paleoseismic records: Stringing pearls. *Bulletin of the Seismological Society of America*, 99(2A), pp.471-498.

Biasi G. P., and Wesnousky S. G. 2021. Rupture passing probabilities at fault bends and steps, with application to rupture length probabilities for earthquake early warning, *Bull. Seismol. Soc. Am.* 111, 2235–2247.

Field, E.H., 2015. Computing elastic-rebound-motivated earthquake probabilities in unsegmented fault models: A new methodology supported by physics-based simulators. *Bulletin of the Seismological Society of America*, 105(2A), pp.544-559.

Field, E.H., Biasi, G.P., Bird, P., Dawson, T.E., Felzer, K.R., Jackson, D.D., Johnson, K.M., Jordan, T.H., Madden, C., Michael, A.J. and Milner, K.R., 2015. Long-term time-dependent probabilities for the third Uniform California Earthquake Rupture Forecast (UCERF3). *Bulletin of the Seismological Society of America*, 105(2A), pp.511-543.

Goh, K.-I., & Barabási, A.-L. (2008). Burstiness and memory in complex systems. *Europhysics Letters*, 81(4), 48002. <https://doi.org/10.1209/0295-5075/81/48002>

Griffin, J.D., Stirling, M.W. and Wang, T., 2020. Periodicity and clustering in the long-term earthquake record. *Geophysical Research Letters*, 47(22), p.e2020GL089272.

Kearse, J., Little, T.A., Van Dissen, R.J., Barnes, P.M., Langridge, R., Mountjoy, J., Ries, W., Villamor, P., Clark, K.J., Benson, A. and Lamarche, G., 2018. Onshore to offshore ground-surface and seabed rupture of the Jordan–Kekerengu–Needles fault network during the 2016 *M* w 7.8 Kaikōura earthquake, New Zealand. *Bulletin of the Seismological Society of America*, 108(3B), pp.1573-1595.

Kempf, P. and Moernaut, J., 2021. Age uncertainty in recurrence analysis of paleoseismic records. *Journal of Geophysical Research: Solid Earth*, 126(8), p.e2021JB021996.

McCalpin, J.P. and Nelson, A.R., 2009. Introduction to paleoseismology. *International Geophysics*, 95, pp.1-27.

Walsh, E., Stahl, T., Howell, A. and Robinson, T., 2023. Two-Dimensional Empirical Rupture Simulation: Examples and Applications to Seismic Hazard for the Kaikōura Region, New Zealand. *Seismological Society of America*, 94(2A), pp.852-870.

Williams, R.T., 2022. Poisson behavior leads to bias when testing for periodicity in the paleoseismic record of large earthquakes. *Seismological Research Letters*, 93(1), pp.118-125.

- Although I am very positive on the "averaging" approach proposed by the authors I think that the authors should modify their terminology and their procedure to assign weights. The Bayesian Model Averaging (BMA) is something different from what is made by the authors, but, more important, BMA is not an averaging tool, but a model selector. This problem has been deeply discussed by Monteith et al. (2011; Monteith, K., J. L. Carroll, K. Seppi, and T. Martinez (2011). Turning Bayesian model averaging into Bayesian model combination, in Proc. of International Joint Conference on Neural Networks, San Jose, California, 31 July–5 August 2011, 2657–2663.).

Reply: We agree that we should have been more careful about our use of the term Bayesian Model Averaging. We have amended the text to make this clearer. In particular

1. We make it clearer that our approach is not to be confused with what we call classical Bayesian Model Averaging (CBMA), the limitations of which have been discussed at length by one of the authors of this paper (Fletcher 2018).
2. We now discuss the connection between WAIC and Bayesian cross validation.

See Lines 429-439 which is also included below.

“To calculate Bayesian model-averaged forecasts, we combine the posterior distributions of the forecast under each model using model weights. We use prediction-based Bayesian model-averaging (PBMA) (Fletcher, 2018) with the model weights calculated using the Watanabe-Akaike Information Criterion (WAIC, Watanabe & Opper, 2010). This is far less sensitive to the priors for the parameters than classical Bayesian model-averaging (CBMA), which uses posterior model probabilities (Fletcher, 2018). PBMA is sometimes referred to as Bayesian model combination (Monteith et al., 2011). Unlike CBMA, it does not involve the assumption that one of the models is true. For large samples PBMA will lead to use of the best linear combination of the posterior distribution of the forecast from each model, whereas CBMA will lead to use of the posterior distribution of the forecast from the single best model (Monteith et al., 2011; Marzocchi et al., 2012; Yao et al., 2018), which is why some authors refer to CBMA as a tool for model selection (Monteith et al., 2011).”

Fletcher, D. (2018). Model averaging. Springer.

This problem has been found and discussed also in seismology (Marzocchi Zechar and Jordan, 2012, BSSA). The way in which the authors calibrate the weights, using a quantity that is proportional to probability (see their equation 8), is similar to BMA and it might have the same problems (see the literature mentioned above).

Reply: We have revised the manuscript to make it clear that our approach will not suffer from the same problems as CBMA. See Lines 429-439 which is also included in the responses to the previous comment.

It might be interesting for the authors to know that more recently, Herrmann and Marzocchi (GJI, 2023) discussed the optimal weighting scheme for an averaging procedure, which is based on a fundamentally different logic. Here I suggest a more careful re-thinking about the weighting scheme and a solid justification of the method used. Maybe just using a quantity

proportional to WAIC instead of using the exponential, it might improve the problem (This is just a suggestion).

Reply: We thank the reviewer for this suggestion. We agree that the approach in Herrmann and Marzocchi (GJI, 2023) is different. We added further discussions around this point in the revised manuscript. See below which is also in the revised text, Lines 448-452.

“WAIC is a prediction-based criterion, analogous to AIC in the non-Bayesian setting, and use of Equation (8) to define the model weights is motivated by the form of AIC weights (Yao et al., 2018). Alternative approaches to prediction-based model averaging in seismology have been proposed by (Marzocchi et al., 2012) and (Herrmann & Marzocchi, 2023). We prefer to make use of WAIC weights because WAIC has the desirable property of being equivalent to Bayesian cross-validation when used for model selection (Vehtari et al., 2017).”

- Again, despite being positive about the method, I am wondering what is the reason for what faults behave differently (they follow different distributions); is there a different physics? Or what else? In my understanding, the Weibull distribution is the best performing on faults which have a longer dataset. I am wondering what happens if we use the Weibull model for all faults. Of course, in some cases, the Weibull model will perform less well than other models, including the average model, but what is the information gain per event? One model can perform worse than another for a small difference. In this case, the differences do not have any practical importance. I suggest the use of some metric, like the information gain per event, to show how much a model performs better than others and discussing the results.

Reply: We did a regression analysis on the single-best model against tectonic regions and faulting styles, but didn't find significant relationship there. It is difficult to draw conclusion on the underlying physics with the current data.

WAIC provides an estimate of the predictive performance of a Bayesian model (Vehtari 2017). If the WAIC-value for model A is smaller than that for model B, we estimate that model A will provide better predictions than model B. WAIC weights provide a numerical comparison of the amount by which model A is better at prediction than B, as they show how much weight should be given to the prediction from each of these models when calculating a model-averaged prediction. Based on the WAIC weights in Figure 1, we can see that the predictive performance of the Weibull model is not uniformly better than the others. For 33 fault segments, the WAIC weight for the Weibull model is close to 0, which suggests that for these fault segments the estimated predictive accuracy of the other models is much better than that of the Weibull model. We have added this discussion in the revised manuscript, Lines 160-167.

We also calculated the information gain per event for the Weibull renewal process with each of the other four models as a reference for each fault, which range between -0.17 and 0.7. Due to the word limit, this is only for the reviewer's interest and was not included in the revision. Although the information gain per event is positive more often than negative, and the absolute values of the negative gain are mostly smaller than that of the positive ones except for when the Gamma renewal process is used as a reference, we get the same conclusion as when using the WAIC weights: the predictive performance of the Weibull model is not uniformly better than the others.

[REDACTED]

- on a minor side, LINE 30 in the Abstract. I never liked too much the word "robust", because it is unclear what it means. At least to me.

Reply: We have revised the abstract to shorten it and clarified the relevant sentence.

Reviewer #3 (Remarks to the Author):

The manuscript by Wang et al presents a Bayesian statistical analysis of a large number of paleoseismic data sets with the apparent goal of developing an improved forecasting method. The central claims of this manuscript, as I see them, are as follows: 1) Bayesian analysis allows uncertainties in earthquake-recurrence model selection to be more fully accounted for in subsequent forecasting; 2) No single model can account for all of the observed paleoseismic sequences, even those from faults with similar kinematics, strain rates, or tectonic settings; 3) Retrospective forecasting shows that a “model averaging” approach to estimating earthquake occurrence is more informative and accurate than any single other model, including the Poisson (i.e. random earthquakes) model; 4) Various segments of the San Andreas fault system have a high likelihood of rupture (i.e. up to 70% chance) over the course of the next 50 years; and 5) the large dataset analyzed here allows a variety of generalizations about the nature of earthquake recurrence as a function of strain rate, kinematics, and/or tectonic setting.

Of these claims, it is my opinion that (1) and (2) are generally well supported by the data and associated arguments. Claim (3), however, is a much more complicated question, and to be honest, I feel that the manuscript on some level over intellectualizes it. There are practical/societal matters that need to be more fully considered (discussed below). Claim (4) is also well supported by the data and associated discussion, but I do not see where it departs appreciably from a variety of other active forecasting methods that have been applied to the San Andreas system. This kind of begs a question (and I do mean this respectfully) of what exactly is gained by using this new approach when compared with other methods? Finally, claim (5) is also generally well supported here, although at least some of the subclaims related to it have been proposed by other authors who need to be cited. Considering aspects of the presentation, the writing is clear and concise. The quality and utility of the illustrations is similarly top notch. My recommendation is that this manuscript be further considered for publication following moderate revisions.

Reply: We thank the reviewer for these comments and we provide detailed responses below.

I could probably summarize many of my concerns with this manuscript using a single question that I would ask the authors to consider: Why is this approach fundamentally and necessarily better than what has been done previously? In this way, we need to realize that earthquake forecasting is not just an interesting scientific exercise. Rather, it has explicit practical goals, which vary depending on who is asking the question. As a society we make important decisions about how and where to distribute finite resources based on such forecasts. It may be most “correct” for a forecasting approach to more fully consider uncertainties in recurrence-model selection, but that only really matters if doing so leads to a demonstrably more accurate forecast. This is the central issue with all earthquake forecasting work - it's going to be a very long time before we really KNOW if these approaches are of any real utility or which are better than others. Retrospective forecasting is certainly one way to get a tiny glimpse at the answer. It's often unsatisfying though, and it is for me in this case as well, in large part because the confidence intervals for the next event are so large (an unfortunate reality of uncertain datasets, as this manuscript often points out). If our confidence interval for the occurrence of the most recent event in the data set spans hundreds or thousands of years, and we apply it to a record with a mean recurrence interval of about 100 years, is it really surprising/revealing that we successfully capture the event within the predicted interval? So respectfully, why is the approach presented here preferred? How would we pitch it to an insurer? A local government? A builders association? The factor that all of these entities are interested in is less a matter of

the binary occurrence of an earthquake within some window of time, but rather the per-year likelihood of a catastrophic event (often compared with the anticipated useful lifetime of a structure). By decreasing the confidence window of occurrence, the method explicitly raises the per year likelihood of an event, and that has implications. I understand this is a manuscript that is focused on science, so maybe these questions do not need to be explicitly answered, but I do think that the subject matter really does demand at least some engagement with the practical matters at hand.

Reply: We are grateful to the reviewer for giving us the opportunity to make clarifications. We have revised the manuscript accordingly to further clarify the aim of our research and thus answer the question “Why is this approach fundamentally and necessarily better than what has been done previously?” Please see Lines 57-64, 88-99, 171-182, 216-218 in the revised manuscript. We also provide more detailed responses below.

National earthquake rupture forecasts for local faults (used for insurance purposes, or for providing advice to local government or builders associations) are normally done by building in not only a statistical earthquake probability model, but also fault models, deformation models and earthquake rate models. Taking UCERF-3 for California earthquake rupture forecasts for example (Field et al., 2015), the fundamental elastic-rebound theory part of the earthquake probability model is a BPT renewal model.

We aim to use a global record to find a more robust approach for the elastic-rebound theory part of the earthquake probability model, that is, a more robust way to model the inter-arrival times which have been frequently modelled using either the BPT renewal process, because of its physical explanation of the earthquake process (Matthews et al., 2002; Nomura et al., 2011; Field et al., 2015), or the best model among a few candidate renewal processes, based on an information criterion (Ogata, 1999; Rhoades et al., 2011). Improving the forecast from this layer of modelling will result in a holistic hazard model that includes fault models, deformation models, earthquake rate models, and earthquake probability models. We use the global data set to demonstrate that there is no universal model and the BPT renewal process is not necessarily the best model. Choosing a different renewal model or choosing to average over several renewal processes may produce a better forecast.

It is also noted that our study attempts to present a globally applicable method, and that in some more well-studied regions such as the San Andreas Fault System it may be appropriate to develop fault-specific approaches such as that presented by Field et al (2015) that build in additional information about fault segmentation and geometry. It is not our intention to repeat such studies. The method developed in our study could be used as an input to a UCERF3 style model, in terms of providing more accurate constraints on the mean recurrence intervals and aperiodicity, but it is beyond the scope of this study to implement a UCERF3 approach.

Toward the point above, it also seems that any manuscript presenting a new approach to earthquake forecasting should provide some detailed comparisons with previous forecasting work. For example, the earthquake occurrence probabilities for various segments of the San Andreas presented here strike me as largely the same as those presented by the Uniform California Earthquake Rupture Forecast. UCERF also includes a time-dependent forecast in addition to a Poisson. I'll grant that the approach here presents a less uncertain forecast than a Poisson alternative whilst also being broadly as accurate (per the retrospective forecast), but what about other time-dependent models such as UCERF? Again respectfully, why is the

approach presented here better (this is not to express doubt that it may be, but rather that it is not clear WHY it necessarily is)?

Reply: We hope our response to the previous question also answers this question here. UCERF-3 consists of four components, namely the fault model, the deformation model, the earthquake rate model, and underlying these three, the earthquake probability model. We aim to use a global record to find a better approach for the earthquake probability model. It is not our aim to build a full hazard model for all the 93 fault segments worldwide. We use the global data set to demonstrate that there is no universal earthquake probability model and the BPT renewal process is not necessarily the best model. Choosing a different renewal model or choosing to average over several renewal processes may produce a better forecast. The method developed in our study could be used as an input to a UCERF3 style model, in terms of providing more accurate constraints on the mean recurrence intervals and aperiodicity.

Finally, the manuscript consistently does a bit of a hit job on the use of the Poisson model in earthquake forecasting. I think that should be dialed back a bit. Yes, I too think the idea that earthquakes would occur randomly in time is a bit rubbish from a mechanics perspective, and it is not supported by the current state of the paleoseismic literature as the authors seem to agree. That being said, there are practical reasons for the adoption of a Poisson model within earthquake forecasting as described in the line referenced comments (i.e. it is a decent choice when the date of the most recent event is not known).

Reply: We agree that Poisson process is still a valuable model. We have added some discussions around this in our revised manuscript. See Lines 333-335 which is also provided below.

“Having said this, the Poisson process may still be a valuable model when limited data are available, which is the case for many fault segments that are not included in this study because they have less than five events in the record.”

In summary, this is interesting and high quality work. Having communicated with some of the authors on the subject of earthquake statistics and forecasting previously, I know that their desire to do good science is genuine. I think this work too will soon be worthy of publication, but given its nature, some considerably more nuanced discussion of why this approach is better than the alternatives is required.

Reply: Thank you. We really appreciate all your comments on our work.

Randy Williams
22 August 2023

Line Referenced Comments

18-20: Interesting question really - does better incorporation of uncertainty lead to a more robust forecast? Depends on what a “robust” forecast is. Seems at least possible that we end up with a forecast that is no more or less accurate than was the case prior to using this Bayesian approach.

Reply: We agree with the reviewer that the use of “robust” is vague here. We have revised the abstract to shorten it and clarified the relevant sentence.

21-22: Again, this is cool, but minimizing the forecast uncertainty is only a useful thing if the forecast itself is accurate. The central dilemma of earthquake forecasting! We'll need decades (if not centuries) to find out.

Reply: Here we are comparing the model-averaging approach against the single best-model approach. When there is some uncertainty as to the best model, model averaging (MA) outperforms a single best-model (BM) primarily in terms of better representing all the uncertainty. This typically means the coverage of intervals is much better with MA, and can be poor with BM.

40-47: Could simplify this section for the novice with some more plaintext terminology. The Poisson process describes events that effectively occur randomly in time. And the antithesis of that for periodic sequences of course.

Reply: We have made changes to the text. See Lines 49-54 which is also included below.

“A renewal process is a statistical model that describes event occurrences in time, treating each new occurrence as a renewal in which the system is reset. It assumes that the time between events (the inter-event time, such as the time between two consecutive earthquake occurrences) are independent and identically distributed. The Poisson point process is a special type of renewal process that describes events effectively occurring randomly in time, and thus does not have any memory of the history.”

57-60: Maybe rephrase to better emphasize the severity of the problem. I'd wager that relatively few of those 90 records actually have anywhere near 20 distinct events. In my experience records with more than 10 events are quite rare. Many have fewer than 5.

Reply: Thank you for this suggestion. We have added more description on the data here to emphasize the problem. See Lines 79-80 which is also included below.

“(90 out of a total of 93 earthquake records in this study have evidence for fewer than 20 earthquakes; 79 records have evidence for no more than 10 earthquakes; and 30 have evidence for only 5 earthquakes)”

60-61: It might, certainly, but does it? Again, we really don't know at this point.

Reply: We agree, which is why we used “may produce erroneous or unrealistically precise forecasts”.

77: First occurrence of (what I assume is) Brownian passage time. Please define.

Reply: Thank you. We have defined it when it first appeared in the revised manuscript.

98-99: See comment above about record length.

Reply: We have now used “only 3 fault segments having more than 20 events, and 14 having more than 10 events” to emphasize the severity of the problem.

108: Perhaps this comment is too “in the weeds” but I feel like we have known for a while that a lognormal distribution of interevent times is not an appropriate model for earthquakes, largely because its hazard function actually decreases through time from some maximum value.

Reply: We thank the reviewer for this comment. Given that lognormal has been considered by many studies in literature for paleoearthquakes, we think it’s useful to include it for this global study. After reading this comment, we doubled checked our code and found an error when calculating the WAIC weight for the lognormal model: it was a mistake when specifying the standard deviation of the lognormal distribution for the calculation of the WAIC weight. We have thus corrected the error and recalculated everything. Although some numbers reported in the results changed, our conclusions do not change, and are better supported by the new calculations.

147-150: Quite interesting though.

Reply: We have now revised this part to add more discussion. See Lines 171-182 which is also included below.

“It is unclear if this is simply an outcome of sampling or due to real differences in recurrence behaviour between the different segments, including how neighbouring fault segments interact. Variability in observed earthquake recurrence behaviour at paleoseismic sites on the San Andreas Fault has been proposed to be (at least partially) due to overlap of ruptures occurring on neighbouring segments (Weldon et al., 2004) and this has been supported by studies using earthquake simulators (Field, 2015). In contrast, relatively strongly quasi-periodic recurrence on the Alpine Fault has been attributed to its geometric simplicity and relative isolation from other faults (Berryman et al., 2012). The persistence (or otherwise) of rupture barriers between segments (Philibosian & Meltzner, 2020) may also be a significant factor controlling the distribution of inter-event times observed on a fault segment. Therefore, because of the limitations of the available data, even within a single well-studied fault system, we cannot use a universal single best model, and it is not clear that one exists.”

169: Could just call this “27” time as likely?

Reply: Thank you. We have changed this.

207-222: Some of this discussion makes one speculate as to whether variations in strain rate or kinematics have an appreciable impact on record length and/or event preservation. Maybe worth a teaser sentence?

Reply: Thank you. We have added in the revised manuscript the following sentence, Lines 282-284.

“With more data becoming available in the future, one could investigate whether variations in strain rate or kinematics have a significant impact on record length and event preservation.”

211-215: Pretty sure that Berryman made a similar argument in the 2011 Alpine fault paleoseismology paper. Should probably give them a citation.

Reply: Thank you. We have now cited Berryman et al., 2012.

251-252: I think this statement needs a bit of justification. Why is having smaller errors necessarily preferable from a hazards forecasting perspective? As stated above, it's only really a good thing if the forecast is accurate.

Reply: Thank you. We have revised this part to reflect this suggestion. See Lines 326-335 which is also included below.

“Even though in some situations with a small number of events in the record, the less informative, more uncertain Poisson-based forecasts seem to cover the true value, the majority of faults with small numbers of events are still better represented by the model-averaging approach (e.g., 23 out of the 30 fault segments which had 4 events in the retrospective forecasts have model-averaged forecasts with much smaller MSE than the Poisson forecasts). It is anticipated that for most hazard modelling purposes the smaller errors associated with the model-averaged forecasts favours their use. Having said this, the Poisson process may still be a valuable model when limited data are available, which is the case for many fault segments that are not included in this study because they have less than five events in the record.”

253: Although I am also in favor of time-dependent approaches, we can probably cut the Poisson model a little bit of slack here. It does have its uses from a hazards perspective, particularly if we have some sense of the average interseismic period length but not a date on the most recent slip event. We see in the retrospective forecast that in a select few cases smaller errors caused the analysis to miss the most recent event.

Reply: Please see the reply above.

235-240: Yes, the Poisson model does yield larger uncertainties and thus more frequent “overlap” with the actual events, but to be fair, the model average approach also yields fairly large uncertainties in an absolute sense. In the example of the San Andreas, if the forecast produces a 95% confidence interval for the next event that is a period of several hundred years, whilst the actual data suggest that a typical recurrence interval is on the order of about 100 years, it is a bit unsurprising that the forecast would capture the event within its uncertainty. Respectfully, I feel that I have to ask what exactly we learn from this? Certainly it could be interpreted to show that your approach isn't fundamentally off base, but why is this necessarily better than all of the other hazards forecasts that are available for the San Andreas (or other fault systems)?

Reply: As mentioned earlier in one of our replies, we aim to use a global record to find a more robust approach for the elastic-rebound theory part of the earthquake probability model, which can then be used to aid development of a better full hazard model like UCERF3 for local faults that includes fault models, deformation models, earthquake rate models, and earthquake probability models.

Also, the bit about the model averaging approach “outperforming” the Poisson is a bit complicated as written. You state that the model averaging approach successfully covered 79 of the true occurrence times, but then state that the model averaging approach outperforms Poisson because 81 of the model averaging results had lower mean squared errors. Does that not then by definition include 2 records for which your retrospective forecast failed (i.e. the CI did not include the true occurrence time)? Strange to count those as successes of the approach.

Reply: We have now added an figure in the supplement and added the following discussion to clarify this. See Lines 337-339 which is also included below.

“For about half of the fault segments (45 of the 93), the MSEs of the forecasts from the Poisson process are more than twice of that from the model-averaged forecasts (Supplementary Fig. 7).”

Finally, please consider comments above regarding the utility of the Poisson process in cases where the date of the most recent event is unknown. There is practical utility in the time-independent approach in some cases, even if we believe that fault behavior is, in fact, time dependent.

Reply: Please see replies above.

251-252: Again, this is a really tricky area of application. Please justify why this is favored, remembering WHY we want to forecast earthquakes accurately in the first place. We could of course be conservative/cautious and assume that any place that has a large fault that moved within the last 50 ka has a very high earthquake risk, but that would not do good things for local economies / insurance / building budgets. Although I do like this manuscript, I feel it has a tendency to treat seismic forecasting as little more than an intellectual exercise by focusing on what is most “correct” about fault behavior and uncertainty minimization. Neither of these factors are readily translated into accuracy/reliability/usefulness of the forecast, although they may be components of them. In a practical sense, its less about correctly predicting an earthquake within some interval of time. Rather, it's about calculating a per year likelihood of a catastrophic event for a particular area. Your approach by definition maximizes that likelihood by minimizing the predicted interval.

Reply: Please see replies above. To reiterate, we aim to use a global record to find a more robust approach for the elastic-rebound theory part of the earthquake probability model, which can then be used to aid development of a better full hazard model like UCERF3 for local faults that includes fault models, deformation models, earthquake rate models, and earthquake probability models. Currently, many studies, including UCERF3, use the BPT for the elastic-rebound theory part of the probability model. Our research found that the BPT renewal process is not necessarily the best model. Choosing a different renewal model or choosing to average over several renewal processes may produce a better forecast

265-266: I did not see a comparison between the model averaging approach and the “single best model” approach in the retrospective forecasting data?

Reply: We have added Supplementary Fig. 8 in the revised supplement to show the relative differences in MSEs between the single best model and the model-averaging approach, and added this discussion in the revised manuscript. See Lines 336-341, which is also included below.

“The MSEs of the retrospective forecasts from the single-best model approach are very close to those from the model-averaging approach for the majority of the fault segments (Supplementary Fig. 8), all within two times relative difference. Model-averaging with WAIC weights is not usually designed to achieve a better MSE than the single best-model approach. However, when there is some uncertainty as to the best model, model-averaging outperforms a single best-model primarily in terms of better representing all the uncertainties.”

267-269: This might be a stronger point to the work that is not being fully appreciated by the line in question. If we are ever to validate whether our hazards forecasts are meaningful (in the absence of waiting a few hundred years to find out), it may be best achieved by looking at the global distribution of forecasts and how often they ring true.

Reply: Thank you for this suggestion. We agree with you that this may be the way to validate hazard forecasts. However, we decide to leave the sentence as it is as we think that statement should be sufficient.

REVIEWER COMMENTS

Reviewer #1 (Remarks to the Author):

The authors have well addressed the reviewers' concerns and comments. The changes and the additional discussion have significantly improved the quality and clarity of the manuscript. However, there still remains one point that should be further addressed to enhance its contribution to the field.

As a response to my comment, the authors have added Supplementary Fig. 4, which compares the normalized inter-event times between the single-best models and concluded that there does not seem to be any characteristic recurrence behaviour for the fault segments that have the same single-best model.

Although the conclusion is not wrong, the differences between the the single-best models should be discussed.

For example, though not mentioned in the revised manuscript, it is clear that the SDs of scaled inter-event times in the two major models, Weibull and lognormal distributions (Supplementary Fig. 4 (a,d)), are much different in their averages.

While the SDs in the Weibull cases are smaller than 0.8 except for one case, more than half of the SDs in the lognormal cases are larger than 0.8.

The authors should discuss this point.

In addition, I guess the reason for this difference might be related to the long tail of the lognormal distribution.

If there is an exceptionally long inter-event time in the records, it solely increases the coefficient of variation and also provides much higher likelihood to the lognormal model than the other models because of its long tail.

I think this might be the case for the lognormal cases with large SDs, and that is why I suggested in the previous round of review that a histogram of the normalised inter-event times be drawn for each single best model.

Reviewer #2 (Remarks to the Author):

I think that the authors address well my main concerns, maybe except one.

I do appreciate the fact that the authors now make a distinction between CBMA and PBMA.

However, I noticed one possible inconsistency; or, at least something that is not clear to me. The authors claim that "For large samples PBMA will lead to use of the best linear combination of the posterior distribution of the forecast from

each model, ..."; however, according to the authors "WAIC weights provide a numerical comparison of the amount by which a model is better at prediction than another, as they show how much weight

should be given to the prediction from each of these models when calculating a model-averaged prediction."

It is not clear to me how using the weights that reflect the forecasting performance of each single model will guarantee, with a large sample, to get the best performance of the linear combination. Of course, there must be a connection and the ranking of the models will be the same (the best performing model will likely have also the largest coefficient in the linear combination), but I think that using the weights that account for the individual performance of each single model does not guarantee that the linear combination would be the best one. If I am wrong (and I could be!), it would be good to add more explanations and/or references that address this issue in depth.

Reviewer #3 (Remarks to the Author):

Thank you for taking the time to address my previous comments. I think in general this manuscript has been improved relative to the original version. I would suggest that a bit of revising be done in the introduction, largely because although useful additional details and potential for clarity have been added, my opinion is that the quality of the writing went down slightly. It reads a bit as though the additional pieces of information requested by the reviewers were added without consideration of overall flow. This is not a huge problem though, and I do not feel it should hold up publication.

The only suggestion I made that was not implemented was to provide some comparisons between the probabilistic forecasting provided by this model (beginning on line 204) and that provided by other approaches (UCERF, etc) where possible. This suggestion was dismissed on the basis that other efforts like UCERF are multi component forecast models, whereas this manuscript seeks to model the renewal / elastic rebound process only. Respectfully, I don't understand this argument. Other models like UCERF provide, amongst other things, probabilistic forecasts of the likelihood of large earthquakes over some forward period of time. This manuscript also provides such forecasts. As such, I do not see why they could not or should not be compared. It is really a pretty basic and fundamental question. I do not necessarily think this manuscript has to include probabilistic forecasts as a result/product, but it does in current form, and as such you have to expect that people will consider them on a case by case basis for an area of interest/concern. A comparison with other forecasts (where available) is warranted. I do not think this would be complicated to do in a qualitative way, and as such my recommendation is that this manuscript be further considered for publication following minor revisions. I think at a minimum, an explicit statement about why that comparison is not included should be provided.

Overall though, I'll look forward to seeing this in print. Best of luck with the remainder of the process.

Randy Williams
20 October 2023

RESPONSES TO REVIEWER COMMENTS

Reviewer #1 (Remarks to the Author):

The authors have well addressed the reviewers' concerns and comments. The changes and the additional discussion have significantly improved the quality and clarity of the manuscript. However, there still remains one point that should be further addressed to enhance its contribution to the field.

As a response to my comment, the authors have added Supplementary Fig. 4, which compares the normalized inter-event times between the single-best models and concluded that there does not seem to be any characteristic recurrence behaviour for the fault segments that have the same single-best model. Although the conclusion is not wrong, the differences between the single-best models should be discussed. For example, though not mentioned in the revised manuscript, it is clear that the SDs of scaled inter-event times in the two major models, Weibull and lognormal distributions (Supplementary Fig. 4 (a,d)), are much different in their averages. While the SDs in the Weibull cases are smaller than 0.8 except for one case, more than half of the SDs in the lognormal cases are larger than 0.8. The authors should discuss this point.

In addition, I guess the reason for this difference might be related to the long tail of the lognormal distribution. If there is an exceptionally long inter-event time in the records, it solely increases the coefficient of variation and also provides much higher likelihood to the lognormal model than the other models because of its long tail. I think this might be the case for the lognormal cases with large SDs, and that is why I suggested in the previous round of review that a histogram of the normalised inter-event times be drawn for each single best model.

Reply: We thank the reviewer for this observation and for the thoughtful suggestions. We have added the suggested discussion about the differences in the median standard deviations of the scaled inter-event times between the fault segments that were best fit by a Weibull model and those that were best fit by a lognormal model in the revised manuscript. We also added some discussion around the reason for this difference. The reviewer is right that this is related to fault segments that have inter-event times showing a long-tailed distribution which is best captured by a lognormal renewal process. The long inter-event times at the tail end increase the spread of the data and thus we see larger standard deviations of the scaled inter-event times for the fault segments that were best fit by a lognormal model. We added this discussion in the revised manuscript as well. See lines 182-194 which is also included below.

“For each of the 93 fault segments, we calculated the scaled inter-event times for each MC sample, and then reported the median and the 2.5% and 97.5% quantiles of the standard deviations of the scaled inter-event times calculated for the 100 MC samples. Fault segments with higher rate of earthquake occurrences appear to have smaller standard deviations of the scaled inter-event times (Supplementary Fig. 1 (e)). The median standard deviations of the scaled inter-event times for the 41 fault segments that were best fit by a Weibull model are all smaller than 0.8 except for one case. In contrast, more than half of the median standard deviations of the scaled inter-event times for the 15 fault segments that were best fit by a lognormal model are larger than 0.8 (Supplementary Fig. 1 (a,d)). The large median standard deviations are related to fault segments that have inter-event times with long-tailed distributions which result in a large spread of data values. These long-tailed distributions are best fit by a lognormal renewal process.”

Reviewer #2 (Remarks to the Author):

I think that the authors address well my main concerns, maybe except one.

I do appreciate the fact that the authors now make a distinction between CBMA and PBMA. However, I noticed one possible inconsistency; or, at least something that is not clear to me. The authors claim that "For large samples PBMA will lead to use of the best linear combination of the posterior distribution of the forecast from each model, ..."; however, according to the authors "WAIC weights provide a numerical comparison of the amount by which a model is better at prediction than another, as they show how much weight should be given to the prediction from each of these models when calculating a model-averaged prediction."

It is not clear to me how using the weights that reflect the forecasting performance of each single model will guarantee, with a large sample, to get the best performance of the linear combination. Of course, there must be a connection and the ranking of the models will be the same (the best performing model will likely have also the largest coefficient in the linear combination), but I think that using the weights that account for the individual performance of each single model does not guarantee that the linear combination would be the best one. If I am wrong (and I could be!), it would be good to add more explanations and/or references that address this issue in depth.

Reply: We are grateful for this comment, as it made us realise that we could have made this point more clearly. In particular, we now state the following on lines 432-442.

"We prefer to make use of WAIC weights for the following reasons. Model selection using WAIC has the desirable property, in large samples, of being equivalent to Bayesian leave-one-out cross-validation (B-LOO)⁴³. When B-LOO is used in model averaging it is known as Bayesian stacking, and has the useful property that, for large samples, it leads to the best linear combination of the posterior distributions of the forecasts from each model, whereas CBMA will lead to use of the posterior distribution of the forecast from the single best model³⁹⁻⁴¹ (which is why some authors refer to CBMA as a tool for model selection³⁹). We would therefore expect WAIC weights to provide a close-to-optimal linear combination of the posterior distributions of the forecasts from each model, whilst being much less computationally-intensive than Bayesian stacking."

Reviewer #3 (Remarks to the Author):

Thank you for taking the time to address my previous comments. I think in general this manuscript has been improved relative to the original version. I would suggest that a bit of revising be done in the introduction, largely because although useful additional details and potential for clarity have been added, my opinion is that the quality of the writing went down slightly. It reads a bit as though the additional pieces of information requested by the reviewers were added without consideration of overall flow. This is not a huge problem though, and I do not feel it should hold up publication.

Reply: Thank you for this suggestion. We have revised the introduction to make it flow better. Please see lines 43-54 and 66-91.

The only suggestion I made that was not implemented was to provide some comparisons between the probabilistic forecasting provided by this model (beginning on line 204) and that provided by other approaches (UCERF, etc) where possible. This suggestion was dismissed on the basis that other efforts like UCERF are multi component forecast models, whereas this manuscript seeks to model the renewal / elastic rebound process only. Respectfully, I don't understand this argument. Other models like UCERF provide, amongst other things, probabilistic forecasts of the likelihood of large earthquakes over some forward period of time. This manuscript also provides such forecasts. As such, I do not see why they could not or should not be compared. It is really a pretty basic and fundamental question. I do not necessarily think this manuscript has to include probabilistic forecasts as a result/product, but it does in current form, and as such you have to expect that people will consider them on a case by case basis for an area of interest/concern. A comparison with other forecasts (where available) is warranted. I do not think this would be complicated to do in a qualitative way, and as such my recommendation is that this manuscript be further considered for publication following minor revisions. I think at a minimum, an explicit statement about why that comparison is not included should be provided.

Reply: Thank you for this suggestion. We agree with the reviewer that a comparison with the forecasts from other studies is important. We did try to compare but found that many publications of the paleoearthquake records only provide an estimate of the mean recurrence interval which is equivalent to a forecast of the next occurrence time using a Poisson process (this comparison is shown in Supplementary Fig. 2-4). For studies that also provide a forecast of the probability of the next earthquake occurrence within a time period, they use different forecast intervals, which make it difficult to provide a meaningful comparison. For example, UCERF3 provides 30 year probabilities from 2014 aggregated by parent fault section, whereas we provide 50 year probabilities from 2022 on each fault segment. For the San Andreas fault segments, UCERF3 gives 30 year $M \geq 6.7$ mean probabilities (min, max):

Big Bend: 18% (0.2%, 69%), Carrizo: 22% (0.1%, 69%), Coachella: 27% (6.7%, 60%), and Mojave South (we used Pallet Ck site): 23% (1.5%, 75%);

our model-averaged forecast gives 50 year median probabilities (95% CI):

Big Bend: 41% (36%, 46%), Carrizo: 60% (51%, 72%), Coachella: 24% (20%, 30%), Pallet Ck: 45% (40%, 50%).

For our forecasts above, the probabilities for the four segments are very different (three of these differences are statistically significant at the 5% level) with Carrizo having the highest probability. However, from UCERF3 forecasts, all four segments seem to have very similar

forecast probabilities, although they use min and max values rather than 95% confidence intervals and so this is not comparing the same intervals.

Despite this, we endeavoured to compile a table of past forecast probabilities from the 93 fault segments wherever possible and put that next to our 50 year forecast probabilities. This is included as Supplementary Table 1 and Supplementary Fig. 5 (included below). We also added the following text in the revised manuscript. See lines 230-244.

“A quantitative comparison of our probabilistic forecasts with those published in other studies is challenging (Supplementary Table 1). Reports of the mean recurrence interval estimates are equivalent to forecasting the next occurrence time using a Poisson process (this comparison is in Supplementary Fig. 2–4). A meaningful comparison is difficult when the forecast of the probability of the next earthquake occurrence is specified in terms of a start date and a fixed time period, as these vary between studies. For example, UCERF3 provides 30 year probabilities from 2014 CE aggregated by parent fault section^[16], whereas we provide 50 year probabilities from 2022 CE on each fault segment. The UCERF3 forecasts for the San Andreas fault segments all have similar mean values and overlapping ranges, whereas our forecast probabilities show greater variability between segments, with several non-overlapping credible intervals (Supplementary Fig. 5). We found previous forecasts for another eight fault segments in our study. All except two of our forecasts overlap within uncertainties with previous values, although our uncertainty bounds are typically narrower (Supplementary Fig. 5). Having said that, UCERF3 provides the ranges of forecast probabilities rather than 95% confidence intervals.”

Supplementary Fig. 5. Comparison of forecast probabilities from our study with that from previous studies. The fault index in the x-axis follows the index in Table 1 in the main manuscript. San Andreas fault segments have indices from 59 to 69. The black dots are median forecast probabilities with the black bars showing 95% credible intervals. The red dots are mean forecast probabilities with the red bars showing either min and max or 95% confidence intervals. See Supplementary Table 1 for more details about the data used in this plot.

Overall though, I'll look forward to seeing this in print. Best of luck with the remainder of the process.

Reply: Thank you for all your constructive suggestions and comments, which have improved our manuscript significantly.

Randy Williams
20 October 2023

REVIEWERS' COMMENTS

Reviewer #1 (Remarks to the Author):

The authors have addressed my concerns well and revised the manuscript accordingly. I believe that the reviewers' concerns have been sufficiently addressed and that the revised manuscript is worthy of publication as it stands.

Reviewer #2 (Remarks to the Author):

Thank you to the authors to have addressed satisfactorily all my main concerns.

Reviewer #3 (Remarks to the Author):

Thank you for considering my comments and addressing my last remaining concern with the manuscript. I agree that the forecast comparisons that you are able to provide are not "apples to apples" in terms of their temporal formulation, but I do think that the comparison is still useful. The revised text does a good job of discussing those issues succinctly. I appreciate your efforts to provide these data, and I have nothing further to add in terms of comments. Good luck with the remainder of the publication process!

Randy Williams
5 December 2023